# Reward-based Autonomous Online Learning Framework for Resilient Cooperative Target Monitoring using a Swarm of Robots

**Shubhankar Gupta**                                          *shubhankarg@iisc.ac.in*
*Artificial Intelligence and Robotics Lab*
*Department of Aerospace Engineering*
*Indian Institute of Science, Bengaluru, India*

**Saksham Sharma**                                          *saksham.prob@gmail.com*
*Artificial Intelligence and Robotics Lab*
*Department of Aerospace Engineering*
*Indian Institute of Science, Bengaluru, India*

**Suresh Sundaram**                                          *vssuresh@iisc.ac.in*
*Artificial Intelligence and Robotics Lab*
*Department of Aerospace Engineering*
*Indian Institute of Science, Bengaluru, India*

**Reviewed on OpenReview:** *https://openreview.net/forum?id=PzmaWLqKOe*

## Abstract

This paper addresses the problem of decentralized cooperative monitoring of an agile target using a swarm of robots undergoing dynamic sensor failures. Each robot is equipped with a proprioceptive sensor suite for the estimation of its own pose and an exteroceptive sensor suite for target detection and position estimation with a limited field of view. Further, the robots use broadcast-based communication modules with a limited communication radius and bandwidth. The uncertainty in the system and the environment can lead to intermittent communication link drops, target visual loss, and large biases in the sensors' estimation output due to temporary or permanent failures. Robotic swarms often operate without leaders, supervisors, or landmarks, i.e., without the availability of ground truth regarding pose information. In such scenarios, each robot is required to exhibit autonomous learning by taking charge of its own learning process while making the most out of available information. In this regard, a novel Autonomous Online Learning (AOL) framework has been proposed, in which a decentralized online learning mechanism driven by reward-like signals, is intertwined with an implicit adaptive consensus-based, two-layered, weighted information fusion process that utilizes the robots' observations and their shared information, thereby ensuring resilience in the robotic swarm. In order to study the effect of loss or reward design in the local and social learning layers, three AOL variants are presented. A novel *perturbation-greedy* reward design is introduced in the learning layers of two variants, leading to exploration-exploitation in their information fusion's weights' space. Convergence analysis of the weights is carried out, showing that the weights converge under reasonable assumptions. Simulation results show that the AOL variant using the perturbation-greedy reward in its local learning layer performs the best, doing 182.2% to 652% and 94.7% to 150.4% better than the baselines in terms of detection score and closeness score per robot, respectively, as the total number of robots is increased from 5 to 30. Further, AOL's Sim2Real implementation has been validated using a ROS-Gazebo setup.

# 1 Introduction

Target search, detection, tracking, and monitoring are crucial elements in a variety of high-impact real-world applications like search and rescue (Scherer et al., 2015), firefighting (Harikumar et al., 2018; Zhang et al., 2022), convoy protection (Spry et al., 2005), traffic monitoring (Khan et al., 2020), surveillance, etc., where the use of a multi-robot system can prove to be advantageous in terms of robustness and tracking performance, especially in unsafe and uncertain environments (Mohiuddin et al., 2020). With the advancement in sensor and communication technologies, making them miniaturized, inexpensive, and reliable, there has been an increased interest in the use of robotic swarms for target search and tracking (Senanayake et al., 2016). Robotic swarms mostly operate under the philosophy of Swarm Intelligence (SI) (Beni & Wang, 1993; Beni, 2004), which is characterized by decentralized local sensing and control, local communication, and the emergence of self-organizing global behaviors (Sharkey, 2007), along with three main advantages – scalability, robustness, and adaptability (Bayindir & Şahin, 2007), which are highly desirable in robotic swarms.

The problem of cooperative target search and tracking has been studied widely in the literature. There are many target-tracking works that solve the multi-robot problem settings involving an agile target being chased by a swarm of robots. However, most of these works tackle this problem mainly from a control perspective. Moreover, such works assume accurate localization as well as target position estimation (Senanayake et al., 2016), i.e., they do not consider temporary or permanent failures in the proprioceptive (IMU, INS, odometry, etc.) and exteroceptive (camera, LiDAR, RaDAR, etc. – algorithmic/sensor uncertainty in detection and relative pose sensing) sensors onboard the robots. Yao et al. (2007) propose a stable and decentralized control strategy based on artificial potentials and sliding mode control to capture a moving target using a robotic swarm in a specific formation, where the artificial potentials take care of both tracking and formation tasks, and the sliding mode control ensures that the robots follow the required motion. Wang & Gu (2011) consider the problem of cooperative target tracking using a robot swarm with limited communication range, and propose a distributed Kalman filter-based estimation scheme with implicit consensus for the target's position estimation and a distributed flocking algorithm for motion control. Blazovics et al. (2012) propose a simple rule-based distributed algorithm for target tracking and surrounding using a swarm of homogeneous robots based on the concept of basis behaviors, where the robots are aware of the target's position at all times. To achieve dynamic obstacle avoidance and target tracking using a swarm of robots, Radian et al. (2013) use a distributed Kalman filter to estimate the velocity and position of the unknown dynamic convex obstacles and a stochastic target, and a potential field approach to enable target tracking and obstacle avoidance among the robots. Kwa et al. (2020) present a fully decentralized swarming strategy offering tunable exploration-exploitation multi-agent dynamics, which is achieved by combining adaptive inter-agent repulsion and an adjustable network PSO-based strategy, thereby resulting in the optimal collective performance of the swarm corresponding to a specific $k$-nearest neighbor graph connectivity.

Unlike many works in the target search and tracking literature, this paper's main focus is on the target's position estimation part of the overall cooperative target search and tracking task, rather than the control part – this paper considers temporary or permanent sensor failures that cause inaccurate localization and target position estimation among the robots in the swarm. The robots use broadcast-based communication modules with a limited communication radius and bandwidth, exteroceptive sensors for target detection and position estimation with limited field-of-view, and proprioceptive sensors for their own position estimation. The robot swarm has to operate under adverse situations involving temporary or permanent failures in the sensors resulting in large biases in their estimates, intermittent communication link drops, and target visual loss. Robot swarms often operate without leaders, supervisors, or landmarks, i.e., without the availability of a ground truth regarding pose information; typically, the robots use the ground truth to identify which sensors or neighboring robots are undergoing failures. Without the availability of a landmark or a leader, the robots are required to collaborate among themselves. In this regard, a decentralized online learning framework called the Autonomous Online Learning (AOL) framework has been proposed in this paper, which is used to drive an adaptive and fault-tolerant information fusion process among robots. The AOL framework takes its philosophical inspiration from the concepts of 'independent learning' (Hockings et al., 2018) and 'self-directed learning' (Garrison, 1997) in educational psychology in which the learner, devoid of

any supervisor or teacher, takes charge of its own learning process while making the most out of the available resources and information.

In the AOL framework, an autonomous online learning mechanism is intertwined with an implicit adaptive consensus-based, two-layered (local and social), weighted information fusion process, which is driven by the robots' observations and their shared information. This enables each robot to figure out which neighboring robot's information about the target's position can be trusted at a given time instant, thus bringing in an aspect of resilience to the robotic swarm – the robots undergoing sensor failures are less likely to be trusted by themselves as well as their neighbors for their sensor information, thereby improving their target's position estimates by giving more weight to the information shared by robots with functional sensors. The learning process in the AOL framework is inspired by that of the exponentially weighted online learning forecaster (Cesa-Bianchi & Lugosi, 2006), which is a centralized online learning algorithm driven by a loss function calculated using ground truth. But unlike the exponentially weighted online learning forecaster, the AOL framework involves decentralized online learning which is driven by reward-like signals, exhibiting exploration-exploitation behaviors in the online learning process. Since the main focus of this paper is not the control part of the target search and tracking task, a simplified target search and tracking control strategy has been used, which can be replaced by any other advanced control strategy. For instance, the control strategies proposed by Radian et al. (2013) and Kwa et al. (2020) can be used along with the AOL framework to achieve better swarm formation control behavior while ensuring resilience in the robotic swarm.

To understand the effect of loss/reward designs in the local and social learning layers, three different AOL algorithms are presented, namely AOL-C, AOL-1P, and AOL-2P. A novel *perturbation-greedy* reward design is introduced in the learning process of two AOL variants, leading to exploration-exploitation in their information fusion's weights' space. Among the AOL variants, AOL-C involves a comparative loss function in its local learning layer, AOL-1P involves the perturbation-greedy reward function in its local learning layer, and AOL-2P involves the perturbation-greedy reward functions in both its local and social learning layers.

Theoretical analysis of the three AOL variants is carried out, showing that the weights converge under reasonable assumptions. The performance of the three AOL variants is then evaluated in a simulated environment with adverse conditions involving sensor failures. The three AOL variants are compared against two decentralized fusion methods – Averaging-Consensus Fusion (ACF) and Kalman-Consensus Fusion (KCF). Averaging consensus and Kalman filter-based approaches have been used in various works (Katragadda et al., 2017; Zhang & Li, 2019; Azam et al., 2020) for the purpose of multi-estimate fusion. Hence, a comparison of the proposed AOL algorithms is provided with ACF and KCF fusion methods as baselines. However, sensor failures inducing sudden biases in the estimates may or may not increase their covariance, and therefore, covariance-based methods may not perform satisfactorily or may even fail. Simulation results show that the best-performing variant, AOL-1P, performs 182.2% to 652% and 94.7% to 150.4% better than the baselines in terms of cumulative average detection score per robot and cumulative average closeness score per robot, respectively, as the total number of robots is increased from 5 to 30. Simulation results further reveal that using the target detection confidence as a reward signal for the update of weights in the social learning layer, as in AOL-1P, does better than using a perturbation-greedy reward-based learning strategy as in AOL-2P. However, using a perturbation-greedy reward-based learning strategy for the update of weights in the local learning layer, as in AOL-1P, does better than using a comparative Euclidean distance-based loss function as in AOL-C. Further, the top two performing AOL variants' (AOL-1P and AOL-C) Sim2Real aspects are evaluated in Gazebo using ROS1.

The rest of this paper is organized as follows: section 2 presents the problem formulation, and Section 3 presents the AOL framework, along with its three variants. Section 4 presents a theoretical analysis of the convergence of weights involved in the AOL variants. Section 5 presents AOL's performance evaluation, comparing the three variants with two baselines. Finally, section 6 concludes this paper. Further, a table of nomenclature is provided in the appendix (A.1), along with a brief discussion on the generalizability of the AOL framework (A.2) and the pseudo-code for all three AOL variants (A.3).

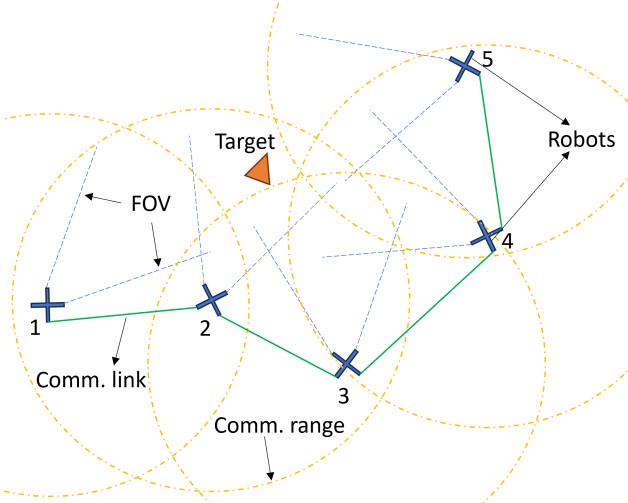

Figure 1: Cooperative Target Monitoring using Robotic Swarm

## 2 Problem Formulation

The problem of Decentralized Cooperative Target Monitoring considered in this paper involves a swarm of robots equipped with broadcast-based communication modules with limited range and bandwidth, proprioceptive sensors (e.g., IMU, INS, optical encoders, etc.), and exteroceptive sensors (e.g., camera, LiDAR, RaDAR, etc.) with limited field-of-view (FOV), as shown in Fig. 1. The goal of the robots is to detect, track, and stay as close as possible to the target while maintaining some distance from it, thus ensuring a high degree of detection and monitoring of the target at all times. However, the sensors onboard the robots may undergo temporary or permanent failures, along with intermittent target visual loss and communication link drops due to system and environmental uncertainty, thereby jeopardizing the target monitoring goal of the robotic swarm. Therefore, the robots are required to collaborate over the communication network to help each other detect and monitor the target successfully.

**Robot Kinematic Model:** let the total no. of robots be $N$. With $\Delta T$ as the sampling period (seconds), each robot $i$, $\forall i \in [N]$, follows a discrete-time 3-DOF kinematic model stated below:

$$\mathbf{x}_{t+1,i} = \mathbf{x}_{t,i} + \Delta T \begin{bmatrix} \cos\phi_{t,i} & -\sin\phi_{t,i} \\ \sin\phi_{t,i} & \cos\phi_{t,i} \end{bmatrix} \bar{\mathbf{v}}_{t,i} \tag{1a}$$

$$\phi_{t+1,i} = \phi_{t,i} + \Delta T \bar{w}_{t,i} \tag{1b}$$

where $\mathbf{x}_{t,i} \in \mathbb{R}^2$ is the $i^{th}$ robot's 2-D position vector (in $m$), $\bar{\mathbf{v}}_{t,i} \in \mathbb{R}^2$ is the $i^{th}$ robot's body-axis velocity vector $(m/s)$, $\phi_{t,i} \in \mathbb{R}$ is the $i^{th}$ robot's heading angle (radians), and $\bar{w}_{t,i} \in \mathbb{R}$ is the $i^{th}$ robot's yaw rate $(rad/s)$ at discrete-time $t$, respectively. Here, the body-axis velocity $\bar{\mathbf{v}}_{t,i}$ and yaw rate $\bar{w}_{t,i}$ are bounded control inputs for the $i^{th}$ robot.

**Target Kinematic Model:** the target's kinematics also follow the model stated as equation (1), but its dynamics is unknown. The target's position vector $\mathbf{x}_{t,B} \in \mathbb{R}^2$ (in $m$), heading angle $\phi_{t,B} \in \mathbb{R}$ (radians), body-axis velocity $\bar{\mathbf{v}}_{t,B} \in \mathbb{R}^2$ (m/s), and yaw rate $\bar{w}_{t,B} \in \mathbb{R}$ $(rad/s)$, respectively, can be represented by replacing $i$ with $B$ (Bogey) in the set of equations (1). Similarly, $\bar{\mathbf{v}}_{t,B}$ and $\bar{w}_{t,B}$ are the bounded control inputs to the target at time $t$, which are unknown to the robots since its dynamics is unknown.

**Communication Model:** let $R_{comm.}$ be the range of communication, and let $p_{ld}$ be the communication link drop probability. The topology of the communication network between robots is represented by a uni-directional dynamic graph $G_t$, whose adjacency matrix $\mathbf{A}_t$ satisfies the following, $\forall i, j \in [N]$ such that $i \neq j$,

$$[\mathbf{A}_t]_{ij} = \begin{cases} 1 & : (\|\mathbf{x}_{t,i} - \mathbf{x}_{t,j}\| \leq R_{comm.}) \wedge (U_{ij}(0,1) \geq p_{ld}) \\ 0 & : otherwise \end{cases} \tag{2}$$

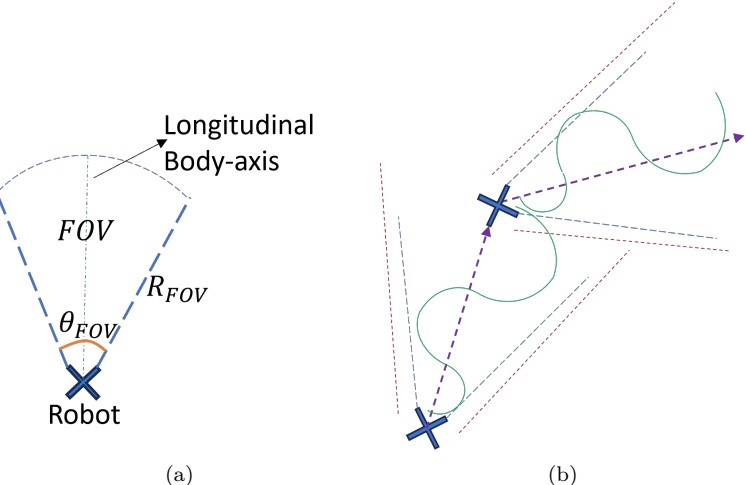

Figure 2: (a) Exteroception's field-of-view (FOV). (b) Target Search strategy inspired by food foraging pattern used by Oxyrrhis Marina.

where $[\mathbf{A}_t]_{ij}$ is the $ij^{th}$ element of $\mathbf{A}_t$, $U_{ij}(0,1) \in [0,1]$ is a uniform random variable. For $i = j$, $[\mathbf{A}_t]_{ii} = 0$. The above equation implies that even if two robots are in the communication range of each other, the communication link between them may still drop with a probability of $p_{ld}$.

The neighbor set of the $i^{th}$ robot at time $t$, denoted by $\Omega_{t,i}$, can be defined as follows, $\forall i \in [N]$:

$$\Omega_{t,i} = \{j \quad : [\mathbf{A}_t]_{ij} = 1 \wedge j \in [N]\} \tag{3}$$

Further, define $n_{t,i} := |\Omega_{t,i}|$, i.e., $n_{t,i}$ is the number of communicating neighbors of the $i^{th}$ robot at time $t$. Moreover, due to limited bandwidth, there is a limit on the number of neighbors the $i^{th}$ robot can have at time $t$, i.e., $n_{t,i} \leq n_l$, where $n_l \in \{1, 2, \cdots, N\}$ based on the limitations posed by the communication hardware. In case more than $n_l$ robots appear in the communication range of the $i^{th}$ robot, $n_l$ robots are chosen randomly out of them to be considered as communication neighbors, such that $n_{t,i} \leq n_l$ holds.

**Proprioceptive Sensor Model:** the proprioceptive sensor suite onboard the $i^{th}$ robot is responsible for the estimation of its 2-D position and yaw angle, where the estimates are denoted as $\hat{\mathbf{x}}_{t,i}^{P_i} \in \mathbb{R}^2$ and $\hat{\phi}_{t,i}^{P_i} \in \mathbb{R}$, respectively, and are modeled as follows:

$$\hat{\mathbf{x}}_{t,i}^{P_i} = \mathbf{x}_{t,i} + \mu_{t,i}^x \tag{4a}$$

$$\hat{\phi}_{t,i}^{P_i} = \phi_{t,i} + \mu_{t,i}^\phi \tag{4b}$$

where $\mu_{t,i}^x \in \mathbb{R}^2$ and $\mu_{t,i}^\phi \in \mathbb{R}$ represent bounded arbitrary noise in the $i^{th}$ robot's proprioceptive sensor suite's estimates, $\forall i \in [N]$.

**Exteroceptive Sensor Model:** the exteroceptive sensor suite of the $i^{th}$ robot is responsible for the detection and relative position estimation of the target, in terms of the target detection confidence $d_{t,i} \in [0,1]$ and the target's relative position estimate $\Delta \hat{\mathbf{x}}_{t,B}^{E_i} \in \mathbb{R}^2$. The exteroceptive sensor suite is further characterized by a limited field-of-view (FOV), with a detection range $R_{FOV}$ and an angle-of-view $\theta_{FOV}$, as shown in Fig. 2a. Let $p_{vl}$ be the probability of target visual loss. If the target lies outside the FOV region, $d_{t,i} = 0$. On the other hand, if the target lies inside the FOV region, the detection confidence $d_{t,i}$ follows the model as stated below:

$$d_{t,i} = \begin{cases} 1 - b_0 \frac{r}{r_0} & : r \leq r_0 \\ (1 - b_0) \exp\left(3 \frac{r_0 - r}{R_{FOV} - r_0}\right) & : r_0 < r \leq R_{FOV} \\ 0 & : (r > R_{FOV}) \vee (U_i(0,1) < p_{vl}) \end{cases} \tag{5}$$

where $r := ||\mathbf{x}_{t,B} - \mathbf{x}_{t,i}||$, $U_i(0,1) \in [0,1]$ is a uniform random variable, $\forall i \in [N]$, and $b_0 \in [0,1)$ and $r_0 \in (0, R_{FOV}]$ are model parameters. The parameters $b_0$ and $r_0$ can be tuned based on the characteristics

of the exteroception system to be installed in the robots. Note that the above model implies a linear decay of the detection confidence till $r = r_0$, after which the decay is exponential, leading to a near zero detection confidence value at $r = R_{FOV}$. In real-world usage, similar behavior is observed in camera and LiDAR-based exteroception (Singh & Davis, 2017; Najibi et al., 2019). Further, even if the target lies in the FOV region of the robot, there may be a visual loss of the target with probability $p_{vl}$.

The relative target position estimation $(\Delta\hat{\mathbf{x}}_{t,B}^{E_i})$ model is stated as follows:

$$\Delta\hat{\mathbf{x}}_{t,B}^{E_i} = \mathbf{x}_{t,B} - \mathbf{x}_{t,i} + \nu_{t,i} \tag{6}$$

where $\nu_{t,i} \in \mathbb{R}^2$ represent bounded arbitrary noise in the $i^{th}$ robot's exteroception's relative target position's estimate, $\forall i \in [N]$.

Thus, the combination of the estimates from proprioception and exteroception yields the combined sensor estimate of the target's position $\hat{\mathbf{x}}_{t,B}^{S_i}$ as follows:

$$\hat{\mathbf{x}}_{t,B}^{S_i} = \hat{\mathbf{x}}_{t,i}^{P_i} + \Delta\hat{\mathbf{x}}_{t,B}^{E_i} \tag{7}$$

**Control Law:** robots use a hybrid switching control law, which switches between the target chase mode and the target search mode based on whether they or their neighbors have detected the target.

For both the modes, the $i^{th}$ robot's translational control law consists of two terms:

$$\bar{\mathbf{v}}_{t,i} = \bar{\mathbf{v}}_{t,i}^R + \Delta\bar{\mathbf{v}}_{t,i} \tag{8}$$

where $\bar{\mathbf{v}}_{t,i}^R$ is the $i^{th}$ robot's velocity reference command signal responsible for either chasing or searching the target and $\Delta\bar{\mathbf{v}}_{t,i}$ is the $i^{th}$ robot's velocity correction control signal responsible for avoiding collisions with other robots (more details in the supplementary document). Note that any other advanced target search and tracking control law can be used here. But for the sake of simplicity, the following search and tracking control strategy has been used in this paper.

*Target Chase Mode:* This mode gets activated whenever the $i^{th}$ robot has detected the target $(d_{t,i} > 0)$ or one of its neighbors has detected the target $(d_{t,j} > 0, j \in \Omega_{t,i})$. In this mode, the $i^{th}$ robot considers its estimate of the target's position $\hat{\mathbf{x}}_{t,B}^i$ as the reference position for the reference velocity control law. The yaw control of the robot makes sure that the robot's longitudinal body axis points towards the target's estimated position $\hat{\mathbf{x}}_{t,B}^i$. A more detailed description, along with the control algorithm, is included in the supplementary document.

*Target Search Mode:* This mode gets activated whenever the $i^{th}$ robot has not detected the target $(d_{t,i} = 0)$ and either none of its neighbors have detected the target $(d_{t,j} = 0, \forall j \in \Omega_{t,i})$ or it has no neighbors $(n_{t,i} = 0)$. In this mode, the $i^{th}$ robot executes a search pattern inspired by the food foraging pattern used by Oxyrrhis Marina (Lowe et al., 2011; Harikumar et al., 2018), as shown in Fig. 2b. The robot first chooses a random direction to move towards. With its longitudinal body axis aligned with that direction, it moves in that direction using its longitudinal velocity control while doing a growing sinusoidal maneuver (green curve in Fig. 2b) using its lateral velocity control to cover more area as it moves. After $T_s$ discrete-time steps, the robot randomly chooses a new direction and repeats the process.

## 3 Reward-based Autonomous Online Learning for Cooperative Target Monitoring

The Reward-based Autonomous Online Learning (AOL) framework proposed in this paper is a decentralized online learning framework designed for cooperative target monitoring using a swarm of robots in a non-stationary environment. Its ability to learn in real-time without the need for any ground truth allows the robots to exhibit high target monitoring success in scenarios where the sensors onboard the robots may be undergoing temporary or permanent failures, even when landmarks, supervisors, or leaders are unavailable. A systems diagram highlighting where the AOL module fits in each of the robots is shown in Fig. 3.

Within the AOL framework, three variants of AOL algorithms are proposed, namely AOL-C, AOL-1P, and AOL-2P, where AOL-C involves a comparative loss function in one of its learning layers, AOL-1P involves a

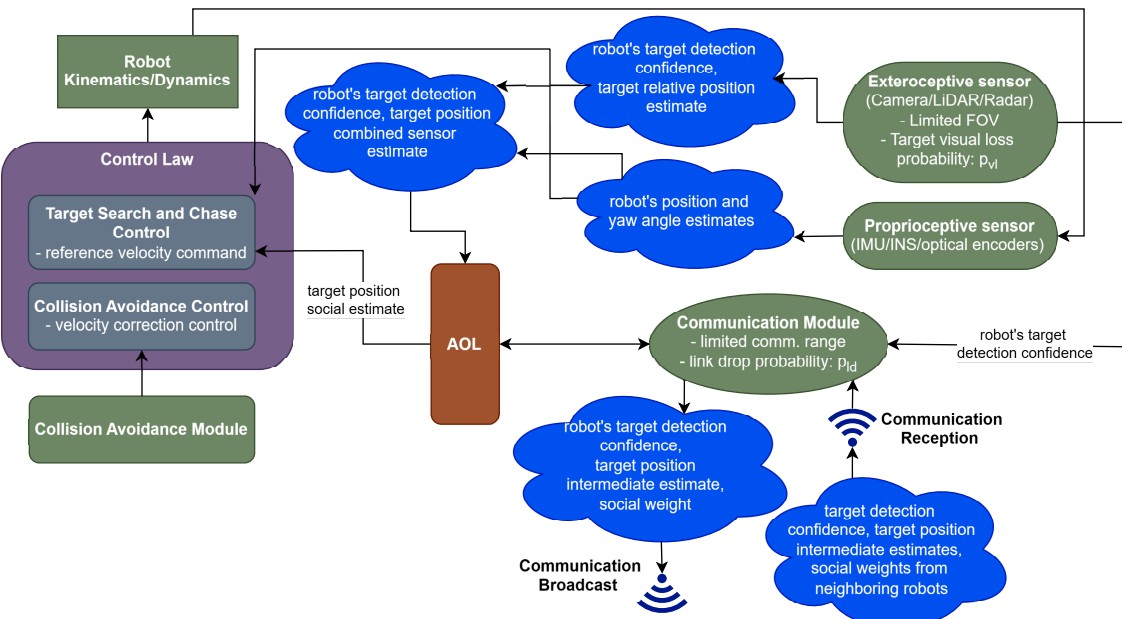

Figure 3: Systems diagram for each robot in the swarm; AOL enables each robot to figure out which neighboring robot's information about the target's position can be trusted at a given time instant, thus bringing in an aspect of resilience to the robotic swarm, especially under sensor failures.

novel *perturbation-greedy* reward function in one of its learning layers, and AOL-2P involves the *perturbation-greedy* reward functions in both of its learning layers. Three different AOL variants are considered in order to study: a) the effect of using comparative loss compared to the perturbation-greedy reward in the local learning layer (AOL-C versus AOL-1P), and b) the effect of using the perturbation-greedy reward in the social learning layer (AOL-1P versus AOL-2P).

All three variants of AOL algorithms involve four phases: a communication phase sandwiched between two estimation phases, and a learning phase. The discrete-time variable $t = 1, 2, \cdots, T$, where $T$ is the discrete-time horizon.

**Local estimation phase:** The first estimation phase involves a weighted fusion of the information available locally to the $i^{th}$ robot, i.e., its combined sensor estimate of the target position $\hat{\mathbf{x}}_{t,B}^{S_i}$ and its previous-time final estimate of the target position $\hat{\mathbf{x}}_{t-1,B}^i$, to form the intermediate estimate $\hat{\mathbf{x}}_{t,B}^{I_i}$ of the target position, as follows:

If the target is detected by $i^{th}$ robot's exteroception, i.e., $d_{t,i} > 0$, then

$$\hat{\mathbf{x}}_{t,B}^{I_i} = \alpha_i(t-1)\hat{\mathbf{x}}_{t,B}^{S_i} + (1 - \alpha_i(t-1))\hat{\mathbf{x}}_{t-1,B}^i \tag{9}$$

otherwise, if the target is undetected by $i^{th}$ robot's exteroception, i.e., $d_{t,i} = 0$, then

$$\hat{\mathbf{x}}_{t,B}^{I_i} = \hat{\mathbf{x}}_{t-1,B}^i \tag{10}$$

Here, the local weight $\alpha_i(t-1)$ is calculated as follows:

$$\alpha_i(t-1) = \frac{\hat{\alpha}_i(t-1)}{\hat{\alpha}_i(t-1) + \hat{\alpha}'_i(t-1)} \tag{11}$$

where the weights $\hat{\alpha}_i(t-1)$ and $\hat{\alpha}'_i(t-1)$ are initialized as $\hat{\alpha}_i(0) = \hat{\alpha}'_i(0) = 1$. These weights are updated in the local learning layer of the learning phase. Note that the weight $\alpha_i(t-1)$ is indicative of how much weight should be given to the combined sensor estimate in the local estimate fusion process at time $t$ relative to the

robot's previous-time final estimate of the target position. Further, let $\hat{\alpha}_i(0) = \hat{\alpha}'_i(0) = \hat{\alpha}_i(-1) = \hat{\alpha}'_i(-1) = 1$ (for the AOL variants with perturbations, called AOL-1P and AOL-2P).

**Communication phase:** The $i^{th}$ robot broadcasts the information $\{t, i, d_{t,i}, \hat{\mathbf{x}}^{I_i}_{t,B}, \hat{\mathbf{x}}^i_{t-1,B}, \hat{w}_{ii}(t-1)\}$ and receives the information $\{t, j, d_{t,j}, \hat{\mathbf{x}}^{I_j}_{t,B}, \hat{\mathbf{x}}^j_{t-1,B}, \hat{w}_{jj}(t-1)\}$ from its communicating neighbors $j \in \Omega_{t,i}$. Here, the weight $\hat{w}_{ii}(t-1)$ is initialized as $\hat{w}_{ii}(0) = 1$, and is updated in the social learning layer of the learning phase.

**Social estimation phase:** The second estimation phase involves a weighted fusion of the information available socially to the robots, i.e., the intermediate estimates $\hat{\mathbf{x}}^{I_j}_{t,B}$ given by its communicating neighbors $j \in \Omega_{t,i}$, and its own intermediate estimate $\hat{\mathbf{x}}^{I_i}_{t,B}$, to form its final estimate $\hat{\mathbf{x}}^i_{t,B}$ of the target's position, as follows:

$$\hat{\mathbf{x}}^i_{t,B} = \sum_{\forall j \in \Lambda_{t,i}} w_{ij}(t-1)\hat{\mathbf{x}}^{I_j}_{t,B} \tag{12}$$

where $\Lambda_{t,i} := \Omega_{t,i} \cup \{i\}$, and the social weights $w_{ij}(t-1)$ are calculated as follows:

$$w_{ij}(t-1) = \frac{\hat{w}_{jj}(t-1)}{\sum_{\forall j' \in \Lambda_{t,i}} \hat{w}_{j'j'}(t-1)} \tag{13}$$

where the weight $\hat{w}_{ii}(t-1)$ is updated in the $i^{th}$ robot's social learning layer of the learning phase, and the weights $\hat{w}_{jj}(t-1)$ are obtained from its communicating neighbors $j \in \Omega_{t,i}$ during the communication phase. The weight $\hat{w}_{ii}(t-1)$ is initialized as $\hat{w}_{ii}(0) = 1, \forall i \in [N]$. Note that the weight $w_{ij}(t-1)$ is indicative of how much weight should be given to the $j^{th}$ robot's intermediate estimate in the social estimate fusion process at time $t$ relative to other robots in the set $\Lambda_{t,i}$. Further, let $\hat{w}_{ii}(0) = \hat{w}_{ii}(-1) = 1$ (for the AOL variants with perturbations, called AOL-1P and AOL-2P).

**Learning phase:** The online learning of the $i^{th}$ robot is primarily driven by the objective of maximizing its target detection confidence $d_{t,i}$ directly or indirectly, which acts as a reward-like signal guiding a multiplicative exponential update process for the weights used in both the estimation phases. The learning phase for all three variants involves a periodic reset of the weights $\hat{\alpha}_i(t)$, $\hat{\alpha}'_i(t)$, and $\hat{w}_{ii}(t)$ to deal with the system uncertainty and non-stationary environment; the periodic reset is required to eliminate any biases accumulated in the learned weights. In practice, the multiplicative exponential weight update process allows for a higher learning rate that can cover the loss in performance due to periodic reset of the learned weights.

Three variants of the AOL learning phase are proposed, resulting in three different AOL algorithms, which are described as follows:

**1) AOL-C:** the weight $\hat{w}_{ii}(t-1)$ is updated via the multiplicative exponential strategy using $l^w_{t,i} := (1 - d_{t,i})$ as the loss function, as follows:

$$\hat{w}_{ii}(t) = \hat{w}_{ii}(t-1)\exp\left(-\eta_w(1 - d_{t,i})\right) \tag{14}$$

where $\eta_w$ is a learning rate parameter. With the above update strategy, note that a decrease in the detection confidence of the $i^{th}$ robot leads to a decrease in the weight $\hat{w}_{ii}(t)$. Equation (14) constitutes as the social learning layer.

The weights $\hat{\alpha}_i(t-1)$ and $\hat{\alpha}'_i(t-1)$ are updated via the multiplicative exponential strategy using the following loss functions

$$l^\alpha_{t,i} := \min\{||\hat{\mathbf{x}}^{S_i}_{t,B} - \hat{\mathbf{x}}^{I_{r^*_{t,i}}}_{t,B}||/D_o, 1\} \tag{15a}$$

$$l^{\alpha'}_{t,i} := \min\{||\hat{\mathbf{x}}^i_{t-1,B} - \hat{\mathbf{x}}^{I_{r^*_{t,i}}}_{t,B}||/D_o, 1\} \tag{15b}$$

where $r^*_{t,i} := \arg\max_{j \in \Lambda_{t,i}} w_{ij}(t-1)$, and $w_{ij}(t-1)$ is calculated as per equation (13). Here, $D_o$ is a normalization parameter, and $r^*_{t,i}{}^{th}$ robot is one whose intermediate estimate is considered to be the most accurate among the estimates of robots in the set $\Lambda_{t,i} = \Omega_{t,i} \cup \{i\}$ based on the weights $w_{ij}(t-1)$. Thus,

the weights $\hat{\alpha}_i(t-1)$ and $\hat{\alpha}'_i(t-1)$ are updated as follows:

$$\hat{\alpha}_i(t) = \hat{\alpha}_i(t-1) \exp\left(-\eta_\alpha l^\alpha_{t,i}\right) \tag{16a}$$

$$\hat{\alpha}'_i(t) = \hat{\alpha}'_i(t-1) \exp\left(-\eta_\alpha l^{\alpha'}_{t,i}\right) \tag{16b}$$

where $\eta_\alpha > 0$ and $\eta_w > 0$ are the learning rate parameters. Equations 16 constitute the local learning layer.

***Comparative Loss Function:*** given by equations (15), the loss functions $l^\alpha_{t,i}$ and $l^{\alpha'}_{t,i}$ are a measure of how close the estimates $\hat{\mathbf{x}}^{S_i}_{t,B}$ and $\hat{\mathbf{x}}^i_{t-1,B}$ are to the socially *best* intermediate estimate $\hat{\mathbf{x}}^{I_{r^*_{t,i}}}_{t,B}$, respectively. Therefore, an increase in these loss functions leads to a decrease in the weights $\hat{\alpha}_i(t)$ and $\hat{\alpha}'_i(t)$, respectively. This implies an increase in the local weight $\alpha_i(t)$ if the estimate $\hat{\mathbf{x}}^{S_i}_{t,B}$ is more close to the socially *best* intermediate estimate compared to $\hat{\mathbf{x}}^i_{t-1,B}$, and vice versa.

Further, the learning phase also involves a periodic reset - the weights $\hat{\alpha}_i(t)$, $\hat{\alpha}'_i(t)$, and $\hat{w}_{ii}(t)$ are reset to 1 after every $T_p$ discrete time steps.

AOL-C is summarized as algorithm 1 (check appendix A.3).

**2) AOL-1P:** update strategy for the weight $\hat{w}_{ii}(t-1)$ is given by equation (14), which is the same as in the previous variant, AOL-C.

The weights $\hat{\alpha}_i(t-1)$ and $\hat{\alpha}'_i(t-1)$ are updated via the multiplicative exponential strategy using a novel *perturbation-greedy* reward function definition described as follows:

$$r^\alpha_{t,i} := e_{a1}\Delta\alpha_i(t-1)\Delta d_i(t) + e_p(1-d_{t,i}) + e_{a2}\alpha_i(t-1)d_{t,i} \tag{17a}$$

$$r^{\alpha'}_{t,i} := -e_{a1}\Delta\alpha_i(t-1)\Delta d_i(t) + e'_p(1-d_{t,i}) + e_{a2}(1-\alpha_i(t-1))d_{t,i} \tag{17b}$$

where $\Delta\alpha_i(t-1) := \alpha_i(t-1) - \alpha_i(t-2)$, $\Delta d_i(t) := d_{t,i} - d_{t-1,i}$, $e_p$ and $e'_p$ are the perturbation signals, and $e_{a1} > 0$ and $e_{a2} > 0$ are learning rate parameters. With the above-described reward function definitions, the weights $\hat{\alpha}_i(t-1)$ and $\hat{\alpha}'_i(t-1)$ are updated as follows:

$$\hat{\alpha}_i(t) = \hat{\alpha}_i(t-1) \exp\left(r^\alpha_{t,i}\right) \tag{18a}$$

$$\hat{\alpha}'_i(t) = \hat{\alpha}'_i(t-1) \exp\left(r^{\alpha'}_{t,i}\right) \tag{18b}$$

where equations (18) constitute the local learning layer.

The learning phase involves a periodic reset - the weights $\hat{\alpha}_i(t)$, $\hat{\alpha}'_i(t)$, and $\hat{w}_{ii}(t)$ are reset to 1 after every $T_p$ discrete time steps, along with the perturbation signal $e_p$ either taking the value $e_p = p_{mag} \in \mathbb{R}_{>0}$ or $e_p = 0$ with equal probability, whereas $e'_p = p_{mag} - e_p$. Otherwise, $e_p = e'_p = 0$ at all other times.

***Perturbation-greedy Reward Function:*** given by equations (17), the reward functions $r^\alpha_{t,i}$ and $r^{\alpha'}_{t,i}$ consist of three terms - a difference-based correction term ($\Delta\alpha_i(t-1)\Delta d_i(t)$), an inertia term ($\alpha_i(t-1)d_{t,i}$ or $(1-\alpha_i(t-1))d_{t,i}$), and a perturbation term ($e_p(1-d_{t,i})$ or $e'_p(1-d_{t,i})$). The difference-based correction term acts as the *greedy* part of the reward function, whereas the role of the inertia term is to resist abrupt changes in the reward due to the *greediness* of the difference-based correction term. The role of the perturbation term is to apply a perturbation periodically, thereby bringing in exploratory behaviors in the online learning process. After a perturbation is applied when the periodic reset is hit, the weight $\alpha_i(t-1)$ may increase ($\Delta\alpha_i(t-1) > 0$) or decrease ($\Delta\alpha_i(t-1) < 0$), possibly leading to a change in the detection performance as well ($\Delta d_i(t) > 0$ or $\Delta d_i(t) < 0$). Therefore, a positive difference-based correction term ($\Delta\alpha_i(t-1)\Delta d_i(t) > 0$) leads to an increase in $r^\alpha_{t,i}$ and a decrease in $r^{\alpha'}_{t,i}$, thereby increasing $\hat{\alpha}_i(t)$ and decreasing $\hat{\alpha}'_i(t)$, and vice versa. Note that the inertia term ($\alpha_i(t-1)d_{t,i}$ or $(1-\alpha_i(t-1))d_{t,i}$) is directly proportional to detection confidence ($d_{t,i}$); a higher detection confidence promotes a larger inertia. Since perturbations should be avoided if the detection confidence is already high, note that the magnitude of the perturbation term ($e_p(1-d_{t,i})$ or $e'_p(1-d_{t,i})$) decreases as the detection confidence $d_{t,i}$ increases.

AOL-1P is summarized as algorithm 2 (check appendix A.3).

**3) AOL-2P:** the weights $\hat{\alpha}_i(t-1)$ and $\hat{\alpha}'_i(t-1)$ are updated using the same strategy as in the previous variant, AOL-1P, given by equations (17) and (18).

The weight $\hat{w}_{ii}(t-1)$ is updated via the multiplicative exponential strategy using a novel *perturbation-greedy* reward function definition described as follows:

$$r_{t,i}^w := e_{w1}\Delta w_{ii}(t-1)\Delta d_i(t) + e_p^{w_i}(1-d_{t,i}) + e_{w2}w_{ii}(t-1)d_{t,i} \tag{19}$$

where $\Delta w_{ii}(t-1) := w_{ii}(t-1) - w_{ii}(t-2)$, $\Delta d_i(t) := d_{t,i} - d_{t-1,i}$, $e_p^{w_i}$ is the perturbation signal, and $e_{w1} > 0$ and $e_{w2} > 0$ are learning rate parameters. The terms $\Delta w_{ii}(t-1)\Delta d_i(t)$ and $w_{ii}(t-1)d_{t,i}$ can be considered as difference-based correction term and the inertia term, respectively, whereas $e_p^{w_i}(1-d_{t,i})$ is the perturbation term that is only active when the periodic reset is hit. With the above-described reward function definition, the weights $\hat{w}_{ii}(t-1)$ are updated as follows:

$$\hat{w}_{ii}(t) = \hat{w}_{ii}(t-1)\exp\left(r_{t,i}^w\right) \tag{20}$$

where equation (20) constitutes the social learning layer.

The learning phase involves a periodic reset - the weights $\hat{\alpha}_i(t)$, $\hat{\alpha}_i'(t)$, and $\hat{w}_{ii}(t)$ are reset to 1 after every $T_p$ discrete time steps, along with the perturbation signal $e_p^{w_i}$ taking the value $e_p^{w_i} = Unif.(0, p_{mag})$, where $p_{mag} \in \mathbb{R}_{>0}$ and $Unif.(0, p_{mag})$ is a uniform random variable; this ensures that the $e_p^{w_i}$ values for different $i \in [N]$ are likely to be different at the periodic reset. Otherwise, $e_p^{w_i} = 0$ at all other times.

***Perturbation-greedy Reward Function:*** given by equation (19), the reward function $r_{t,i}^w$ consists of three terms - a difference-based correction term ($\Delta w_{ii}(t-1)\Delta d_i(t)$), an inertia term ($w_{ii}(t-1)d_{t,i}$), and a perturbation term ($e_p^{w_i}(1-d_{t,i})$). The difference-based correction term acts as the *greedy* part of the reward function, whereas the role of the inertia term is to resist abrupt changes in the reward due to the *greediness* of the difference-based correction term. The role of the perturbation term is to apply a perturbation periodically, thereby bringing in exploratory behaviors in the online learning process. After a perturbation is applied when the periodic reset is hit, the weight $w_{ii}(t-1)$ may increase ($\Delta w_{ii}(t-1) > 0$) or decrease ($\Delta w_{ii}(t-1) < 0$), possibly leading to a change in the detection performance as well ($\Delta d_i(t) > 0$ or $\Delta d_i(t) < 0$). Therefore, a positive difference-based correction term ($\Delta w_{ii}(t-1)\Delta d_i(t) > 0$) leads to an increase in $r_{t,i}^w$, thereby increasing $\hat{w}_{ii}(t)$, and vice versa. Note that the inertia term ($w_{ii}(t-1)d_{t,i}$) is directly proportional to detection confidence ($d_{t,i}$); a higher detection confidence promotes a larger inertia. Since perturbations should be avoided if the detection confidence is already high, note that the magnitude of the perturbation term ($e_p^{w_i}(1-d_{t,i})$) decreases as the detection confidence $d_{t,i}$ increases.

AOL-2P is summarized as algorithm 3 (check appendix A.3).

## 4   Convergence Analysis of Weights

Note that at time $t = 0$, the weights $\hat{\alpha}_i(0) = 1$, $\hat{\alpha}_i'(0) = 1$, and $\hat{w}_{ii}(0) = 1$. Without the loss of generality, an analysis of the three AOL variants is carried out without considering a periodic reset; this is done to gain insights into their convergence behavior just after a periodic reset is hit and just before the next periodic reset is about to hit. Note that each time instant when the periodic reset hits can be considered as $t = 0$ from an analysis perspective. Therefore, the analysis can be regarded as valid for each time interval between two successive periodic reset hits. In practice, given a periodic reset time $T_p$, a higher learning rate (set by learning rate parameters) would lead to quick convergence of the weights before the next periodic reset hits. If the learning rate is low, the weights may not converge to their respective convergence values by the time the periodic reset hits. However, their convergence behavior would remain the same, as shown in the theoretical analysis in this section.

In order to analyze the effect of perturbations in the rewards of AOL-1P and AOL-2P, for ease in theoretical analysis, it is assumed that their reward functions undergo perturbations only at $t = 1$ due to the perturbation-greedy reward design. Note that in practice, the perturbations occur just after the periodic reset after every $T_p$ discrete-time steps.

### 4.1 AOL-C

Consider the loss function $l_{t,i}^w := (1 - d_{t,i})$, and note that $l_{t,i}^w \in [0,1]$, since $d_{t,i} \in [0,1]$. Thus, using equation (14), the weight $\hat{w}_{ii}(t)$ can be written as: $\hat{w}_{ii}(t) = exp(-\eta_w L_{t,i}^w)$, where the cumulative loss $L_{t,i}^w := \sum_{s=1}^t l_{s,i}^w$, and $\hat{w}_{ii}(0) = 1$, $\forall i \in [N]$.

Without the loss of generality, define $j_{t,i}^* := \arg\min_{j \in \Lambda_{t,i}} L_{t,j}^w$, such that the $j_{t,i}^{*}{}^{th}$ robot is a unique robot that incurs the least cumulative loss among the robots in the set $\Lambda_{t,i} = \Omega_{t,i} \cup \{i\}$ at time $t$, where $\Omega_{t,i}$ is the neighbor-set of the $i^{th}$ robot at time $t$.

*Assumption 1:* $\lim_{t \to \infty} \Lambda_{t,i}$ and $\lim_{t \to \infty} j_{t,i}^*$ exist uniquely, such that $\lim_{t \to \infty} \Lambda_{t,i} = \Lambda_{\infty,i}$ and $\lim_{t \to \infty} j_{t,i}^* = j_{\infty,i}^*$, $\forall i \in [N]$.

*Remark 1:* Assumption 1 implies that the neighborhood configuration (in terms of the set $\Lambda_{t,i}$) and the performance configuration (in terms of detection confidence $d_{t,i}$) get fixed as $t \to \infty$. That is, for the $i^{th}$ robot, $\forall i \in [N]$, at $t \to \infty$, there exists a unique robot $j_{\infty,i}^*$ that incurs the least cumulative loss $L_{t,j}^w$ among the robots $j \in \Lambda_{\infty,i}$.

Note that the cumulative loss satisfies $0 \le L_{t,j}^w \le t$ (due to loss function's definition), $\forall j \in \Lambda_{t,i}$, and $L_{t,j_{t,i}^*}^w < L_{t,j}^w$ (due to $j_{t,i}^*$'s definition), $\forall j \in \Lambda_{t,i} \setminus \{j_{t,i}^*\}$.

*Assumption 2:* $L_{t,j}^w - L_{t,j_{t,i}^*}^w \ge \epsilon t^\beta > 0$, such that $\beta \in (0,1]$ and $0 < \epsilon \ll 1$.

*Remark 2:* Assumption 2 implies that the lower bound on the difference between the cumulative loss incurred by the $j^{th}$ robot and that of the $j_{t,i}^{*}{}^{th}$ robot grows sub-linearly ($0 < \beta < 1$) or linearly ($\beta = 1$) with the discrete-time $t$, such that the magnitude and the rate of growth are finite but can be arbitrarily small ($0 < \epsilon \ll 1$ and $0 < \beta \ll 1$). Note that assumption 2 generalizes over many practical scenarios. For instance, consider the scenario where the $j_{t,i}^{*}{}^{th}$ robot is fixed over time, i.e., $j_{t,i}^* = j_{0,i}^*$, and satisfies $l_{t,j}^w - l_{t,j_{0,i}^*}^w \ge \epsilon > 0$, for $t \ge 1$, i.e., the $j_{0,i}^{*}{}^{th}$ robot incurs the least loss at any time $t$. This implies that $L_{t,j}^w - L_{t,j_{0,i}^*}^w \ge \epsilon t > 0$, where $0 < \epsilon \ll 1$, which is a special case under assumption 2 when $\beta = 1$. In practice, assumption 2 would be satisfied for any such scenario where the best ($j_{t,i}^{*}{}^{th}$) robot stays fixed for some finite time duration and may change intermittently over time. In the simulation studies (section 5), such intermittent changes result from temporary/permanent sensor failures, intermittent communication link loss, and target visual loss.

**Theorem 1.** *Under assumptions 1 and 2, $\forall i \in [N]$, AOL-C algorithm's weights $w_{ij}(t)$ satisfy the following:*

$$\lim_{t \to \infty} w_{ij}(t) = 0, \quad \forall j \in \Lambda_{\infty,i} \setminus \{j_{\infty,i}^*\} \tag{21}$$

*and*

$$\lim_{t \to \infty} w_{ij_{\infty,i}^*}(t) = 1 \tag{22}$$

*where $j_{\infty,i}^*$ is the index of the robot that incurs the least cumulative loss among the robots in the set $\Lambda_{\infty,i}$ as $t \to \infty$.*

*Proof.* Since $\hat{w}_{ii}(t) = exp(-\eta_w L_{t,i}^w)$, from equation (13), note that $w_{ij}(t)$ can be re-written as follows:

$$w_{ij}(t) = \frac{exp(-\eta_w L_{t,j}^w)}{\sum_{\forall j' \in \Lambda_{t,i}} exp(-\eta_w L_{t,j'}^w)} \tag{23}$$

Multiply both numerator and denominator by $exp\left(-\eta_w L_{t,j_{t,i}^*}^w\right)$ to get:

$$w_{ij}(t) = \frac{exp(-\eta_w (L_{t,j}^w - L_{t,j_{t,i}^*}^w))}{1 + \sum_{\forall j' \in \Lambda_{t,i} \setminus \{j_{t,i}^*\}} exp(-\eta_w (L_{t,j'}^w - L_{t,j_{t,i}^*}^w))} \tag{24}$$

Note that the cumulative loss satisfies $0 \le L_{t,j}^w \le t$ (due to loss function's definition), $\forall j \in \Lambda_{t,i}$. Using this condition along with assumption 2, $\forall j \in \Lambda_{t,i} \setminus \{j_{t,i}^*\}$, we get

$$t \ge L_{t,j}^w - L_{t,j_{t,i}^*}^w \ge \epsilon t^\beta > 0 \tag{25}$$

where $\beta \in (0,1]$ and $0 < \epsilon \ll 1$. Using equation (25) in equation (24), $\forall j \in \Lambda_{t,i} \setminus \{j_{t,i}^*\}$, we get

$$\frac{exp(-\eta_w t)}{1 + \sum_{\forall j' \in \Lambda_{t,i} \setminus \{j_{t,i}^*\}} exp(-\eta_w \epsilon t^\beta)} \le w_{ij}(t) \le \frac{exp(-\eta_w \epsilon t^\beta)}{1 + \sum_{\forall j' \in \Lambda_{t,i} \setminus \{j_{t,i}^*\}} exp(-\eta_w t)} \tag{26}$$

and for $j = j_{t,i}^*$, we get

$$\frac{1}{1 + \sum_{\forall j' \in \Lambda_{t,i} \setminus \{j_{t,i}^*\}} exp(-\eta_w \epsilon t^\beta)} \le w_{ij_{t,i}^*}(t) \le \frac{1}{1 + \sum_{\forall j' \in \Lambda_{t,i} \setminus \{j_{t,i}^*\}} exp(-\eta_w t)} \tag{27}$$

Taking $\lim_{t \to \infty}(\cdot)$ on both sides in equations (26) and (27), under assumption 1, gives the desired result as equations (21) and (22). $\qquad\square$

Considering the loss functions given by equations (15a) and (15b), note that $l_{t,i}^\alpha \in [0,1]$ and $l_{t,i}^{\alpha'} \in [0,1]$. Therefore, the convergence results for the weights $\alpha_i(t)$ can be derived by carrying out an analysis similar to that of the weights $w_{ij}(t)$ under an assumption similar to assumption 2, as shown above.

## 4.2 AOL-1P

Since the update strategy for the weights $\hat{w}_{ii}(t)$ for AOL-1P is the same as that of AOL-C, therefore, the convergence proof for AOL-1P's weights $w_{ij}(t)$ is given by theorem 1.

Consider the weights $\alpha_i(t)$, $\forall i \in [N]$, defined by equation (11), and the update strategy given by equations (17) and (18). Define $R_{t,i}^\alpha := \sum_{s=1}^t r_{t,i}^\alpha$ and $R_{t,i}^{\alpha'} := \sum_{s=1}^t r_{t,i}^{\alpha'}$, where $r_{t,i}^\alpha$ and $r_{t,i}^{\alpha'}$ are given by equations (17a) and (17b), respectively. Note that $\hat{\alpha}_i(0) = 1$ and $\hat{\alpha}_i'(0) = 1$. Further, $\alpha_i(t) \in [0,1]$, and $d_{t,i} \in [0,1]$, $\forall i \in [N]$. Therefore, for $t \ge 1$, the difference-based correction term satisfies $\Delta\alpha_i(t-1)\Delta d_i(t) \in [-1,1]$, where $\Delta\alpha_i(t-1) = \alpha_i(t-1) - \alpha_i(t-2)$ and $\Delta d_i(t) = d_{t,i} - d_{t-1,i}$.

*Assumption 3:* For $t \ge 1$, the difference-based correction term $\Delta\alpha_i(t-1)\Delta d_i(t)$ satisfies the following: either $\sum_{s=1}^t \Delta\alpha_i(s-1)\Delta d_i(s) \ge \epsilon t^\beta > 0$ or $\sum_{s=1}^t \Delta\alpha_i(s-1)\Delta d_i(s) \le -\epsilon t^\beta < 0$, where $0 < \epsilon \ll 1$, $\beta \in (0,1]$, $\Delta\alpha_i(t-1) = \alpha_i(t-1) - \alpha_i(t-2)$, and $\Delta d_i(t) = d_{t,i} - d_{t-1,i}$.

*Remark 3:* Assumption 3 implies that the cumulative sum of the difference-based terms over time is either lower bounded by a positive term which is linear ($\beta = 1$) or sub-linear ($\beta \in (0,1)$) in time, or upper bounded by a negative term which is linear ($\beta = 1$) or sub-linear ($\beta \in (0,1)$) in time, such that the magnitude and the rate of growth of these bounds are finite but can be arbitrarily small ($0 < \epsilon \ll 1$ and $0 < \beta \ll 1$). Note that assumption 3 generalizes over many practical scenarios. For instance, consider the scenario in which either $\Delta\alpha_i(t-1)\Delta d_i(t) \ge \epsilon > 0$ or $\Delta\alpha_i(t-1)\Delta d_i(t) \le -\epsilon < 0$, for $t \ge 1$, implying either $\sum_{s=1}^t \Delta\alpha_i(s-1)\Delta d_i(s) \ge \epsilon t > 0$ or $\sum_{s=1}^t \Delta\alpha_i(s-1)\Delta d_i(s) \le -\epsilon t < 0$, which is a special case under assumption 3 when $\beta = 1$. This case corresponds to the scenario in which an increase in $\alpha_i(t)$ always causes either an increase or a decrease in $d_{t,i}$ for $t \ge 1$. Considering equation (9), this further implies that either the estimate $\hat{\mathbf{x}}_{t,B}^{S_i}$ or the estimate $\hat{\mathbf{x}}_{t-1,B}^i$ is more accurately estimating the target position $\mathbf{x}_{t,B}$ for $t \ge 1$. In practice, assumption 3 would be satisfied for any such scenario where either $\hat{\mathbf{x}}_{t,B}^{S_i}$ or $\hat{\mathbf{x}}_{t-1,B}^i$ stays more accurate than the other for some finite time duration and but the accuracy ranking among these may change intermittently over time. In the simulation studies (section 5), such intermittent changes result from temporary/permanent sensor failures, intermittent communication link loss, and target visual loss.

**Theorem 2.** *For $t \ge 1$, under assumption 3, given $e_{a2}t^{1-\beta} < 2\epsilon e_{a1}$ (check equations (17)), $\forall i \in [N]$, AOL-1P algorithm's weights $\alpha_i(t)$ satisfy the following:*

$$\lim_{t \to \infty} \alpha_i(t) = 1, \quad if \quad \sum_{s=1}^t \Delta\alpha_i(s-1)\Delta d_i(s) \ge \epsilon t^\beta > 0 \tag{28}$$

*and*

$$\lim_{t \to \infty} \alpha_i(t) = 0, \quad if \quad \sum_{s=1}^t \Delta\alpha_i(s-1)\Delta d_i(s) \le -\epsilon t^\beta < 0 \tag{29}$$

*where $0 < \epsilon \ll 1$, $\beta \in (0,1]$, $\Delta\alpha_i(t-1) = \alpha_i(t-1) - \alpha_i(t-2)$, and $\Delta d_i(t) = d_{t,i} - d_{t-1,i}$.*

*Remark 4:* Since $0 < \epsilon \ll 1$, the condition $e_{a2}t^{1-\beta} < 2\epsilon e_{a1}$ further yields $e_{a2}t^{1-\beta} \ll 2e_{a1}$. This can be satisfied either by choosing $e_{a2} = 0$, or by choosing a time-varying $e_{a2}$ such that $e_{a2}(t) \propto t^{-c}$ for $c > 0$, where $1 - \beta - c \leq 0$. In practice, since the AOL-1P algorithm involves a periodic reset with a period of $T_p$ discrete-time steps, this condition implies $e_{a2}T_p^{1-\beta} \ll 2e_{a1}$. This condition can be satisfied when $e_{a2}$ is chosen to be much smaller than $e_{a1}$ or $e_{a2} \approx 0$.

*Proof.* Note that $\alpha_i(t) \in [0,1]$ and $d_{t,i} \in [0,1]$. Thus, $\Delta\alpha_i(t-1) \in [-1,1]$ and $\Delta d_i(t) \in [-1,1]$. Further, $\alpha_i(t-1)d_{t,i} \in [0,1]$. Using equations (11) and (18), the weights $\alpha_i(t)$, $\forall \in [N]$, can be written as follows

$$\alpha_i(t) = \frac{\exp\left(R_{t,i}^{\alpha}\right)}{\exp\left(R_{t,i}^{\alpha}\right) + \exp\left(R_{t,i}^{\alpha'}\right)} = \frac{1}{1 + \exp\left(R_{t,i}^{\alpha'} - R_{t,i}^{\alpha}\right)} \tag{30}$$

Since the perturbations occur just after $t = 0$, therefore at $t = 1$, either $e_p = p_{mag}$ & $e'_p = 0$ or $e_p = 0$ & $e'_p = p_{mag}$, and for $t > 1$, $e_p = e'_p = 0$ (assumed for ease in analysis, without the loss of generality), $\forall i \in [N]$. Thus, $R_{t,i}^{\alpha'} - R_{t,i}^{\alpha}$ can be written as

$$R_{t,i}^{\alpha'} - R_{t,i}^{\alpha} = \pm p_{mag}(1 - d_{1,i}) - 2e_{a1}\sum_{s=1}^{t}\Delta\alpha_i(s-1)\Delta d_i(s) + e_{a2}\sum_{s=1}^{t}(1 - 2\alpha_i(s-1))d_{s,i} \tag{31}$$

Note that $(1 - 2\alpha_i(t-1))d_{t,i} \in [-1,1]$ and $\Delta\alpha_i(t-1)\Delta d_i(t) \in [-1,1]$. For $t \geq 1$, using assumption 3, we get the following two cases ($0 < \epsilon \ll 1$, $\beta \in (0,1]$):

*1) Case A:* $\sum_{s=1}^{t}\Delta\alpha_i(s-1)\Delta d_i(s) \geq \epsilon t^{\beta} > 0$

Since $\Delta\alpha_i(t-1)\Delta d_i(t) \in [-1,1]$, note that $t \geq \sum_{s=1}^{t}\Delta\alpha_i(s-1)\Delta d_i(s) \geq \epsilon t^{\beta}$. Further, $(1 - 2\alpha_i(t-1))d_{t,i} \in [-1,1]$. Therefore, using equation (31), the weight $\alpha_i(t)$ satisfies the following

$$\frac{1}{1 + \exp\left(\pm p_{mag}(1 - d_{1,i}) - 2\epsilon e_{a1}t^{\beta} + e_{a2}t\right)} \leq \alpha_i(t) \leq \frac{1}{1 + \exp\left(\pm p_{mag}(1 - d_{1,i}) - 2e_{a1}t - e_{a2}t\right)} \tag{32}$$

Taking $\lim_{t\to\infty}(\cdot)$ on equation (32), given $e_{a2}t^{1-\beta} < 2\epsilon e_{a1}$, yields the desired result as equation (28).

*2) Case B:* $\sum_{s=1}^{t}\Delta\alpha_i(s-1)\Delta d_i(s) \leq -\epsilon t^{\beta} < 0$

Since $\Delta\alpha_i(t-1)\Delta d_i(t) \in [-1,1]$, note that $-t \leq \sum_{s=1}^{t}\Delta\alpha_i(s-1)\Delta d_i(s) \leq -\epsilon t^{\beta}$. Further, $(1 - 2\alpha_i(t-1))d_{t,i} \in [-1,1]$. Therefore, using equation (31), the weight $\alpha_i(t)$ satisfies the following

$$\frac{1}{1 + \exp\left(\pm p_{mag}(1 - d_{1,i}) + 2e_{a1}t + e_{a2}t\right)} \leq \alpha_i(t) \leq \frac{1}{1 + \exp\left(\pm p_{mag}(1 - d_{1,i}) + 2\epsilon e_{a1}t^{\beta} - e_{a2}t\right)} \tag{33}$$

Taking $\lim_{t\to\infty}(\cdot)$ on equation (33), given $e_{a2}t^{1-\beta} < 2\epsilon e_{a1}$, yields the desired result as equation (29). $\square$

### 4.3 AOL-2P

Since the update strategy for the weights $\hat{\alpha}_i(t)$ and $\hat{\alpha}'_i(t)$ for AOL-2P is the same as that of AOL-1P, therefore, the convergence proof for AOL-2P's weights $\alpha_i(t)$ is given by theorem 2.

Consider the weights $w_{ii}(t)$, $\forall i \in [N]$, defined by equation (13), and the update strategy given by equations (19) and (20). Define $R_{t,i}^{w} := \sum_{s=1}^{t} r_{t,i}^{w}$, where $r_{t,i}^{w}$ is given by equation (19). Note that $\hat{w}_{ii}(0) = 1$. Further, $w_{ii}(t) \in [0,1]$ and $d_{t,i} \in [0,1]$, $\forall i \in [N]$. Therefore, for $t \geq 1$, the difference-based correction term satisfies $\Delta w_{ii}(t-1)\Delta d_i(t) \in [-1,1]$, where $\Delta w_{ii}(t-1) = w_{ii}(t-1) - w_{ii}(t-2)$ and $\Delta d_i(t) = d_{t,i} - d_{t-1,i}$.

Without the loss of generality, define $q_{t,i}^* := \arg\max_{j \in \Lambda_{t,i}} \sum_{s=1}^{t}\Delta w_{jj}(s-1)\Delta d_j(s)$, such that $q_{t,i}^*$ is unique among $j \in \Lambda_{t,i}$; this implies that a cumulative change in the $q_{t,i}^{*}{}^{th}$ robot's social weight causes the most positively or least negatively aligned cumulative change in its confidence score compared to other robots including the $i^{th}$ robot and its neighbors at time $t$. Therefore, as per the weight update rule given by equations (19) and (20), $q_{t,i}^{*}{}^{th}$ robot is a unique robot among the robots in the set $\Lambda_{t,i}$ whose intermediate estimate is given more weight while performing the weighted fusion as given by equation (12).

*Assumption 4:* $\lim_{t\to\infty} \Lambda_{t,i}$ and $\lim_{t\to\infty} q_{t,i}^*$ exist uniquely, such that $\lim_{t\to\infty} \Lambda_{t,i} = \Lambda_{\infty,i}$ and $\lim_{t\to\infty} q_{t,i}^* = q_{\infty,i}^*$, $\forall i \in [N]$.

*Remark 5:* Assumption 4 implies that the neighborhood configuration (in terms of the set $\Lambda_{t,i}$) and the performance configuration (in terms of the difference-based correction term $(\Delta w_{jj}(t-1)\Delta d_j(t))$) get fixed as $t \to \infty$. That is, for the $i^{th}$ robot, $\forall i \in [N]$, at $t \to \infty$, there exists a unique robot $q_{\infty,i}^*$ that exhibits the greatest cumulative sum of the difference-based correction term (i.e., $\sum_{s=1}^{t} \Delta w_{jj}(s-1)\Delta d_j(s)$) among the robots $j \in \Lambda_{\infty,i}$.

Note that the cumulative sum of difference-based correction terms satisfies $-t \leq \sum_{s=1}^{t} \Delta w_{jj}(s-1)\Delta d_j(s) \leq t$, since $\Delta w_{ii}(t-1)\Delta d_i(t) \in [-1,1]$. Further, due to $q_{t,i}^*$'s definition, the cumulative sum of difference-based correction terms satisfies $\sum_{s=1}^{t} \Delta w_{qq}(s-1)\Delta d_q(s) < \sum_{s=1}^{t} \Delta w_{q_{t,i}^* q_{t,i}^*}(s-1)\Delta d_{q_{t,i}^*}(s)$, $\forall q \in \Lambda_{t,i} \setminus \{q_{t,i}^*\}$.

*Assumption 5:* $\sum_{s=1}^{t} \Delta w_{q_{t,i}^* q_{t,i}^*}(s-1)\Delta d_{q_{t,i}^*}(s) - \sum_{s=1}^{t} \Delta w_{qq}(s-1)\Delta d_q(s) \geq \epsilon t^\beta > 0$, such that $\beta \in (0,1]$ and $0 < \epsilon \ll 1$, $\forall q \in \Lambda_{t,i} \setminus \{q_{t,i}^*\}$.

*Remark 6:* Assumption 5 implies that the difference between the cumulative sum of difference-based correction terms corresponding to $q_{t,i}^*$ and $q$ ($\forall q \in \Lambda_{t,i} \setminus \{q_{t,i}^*\}$), i.e., the difference $\sum_{s=1}^{t} \Delta w_{q_{t,i}^* q_{t,i}^*}(s-1)\Delta d_{q_{t,i}^*}(s) - \sum_{s=1}^{t} \Delta w_{qq}(s-1)\Delta d_q(s)$, has a lower bound that varies linearly ($\beta = 1$) or sub-linearly ($0 < \beta < 1$) with the discrete-time $t$, such that the magnitude and the rate of growth of this lower bound is finite but can be arbitrarily small ($0 < \epsilon \ll 1$ and $0 < \beta \ll 1$). Note that assumption 5 generalizes over many practical scenarios. For instance, consider the scenario where the $q_{t,i}^*{}^{th}$ robot is fixed over time, i.e., $q_{t,i}^* = q_{0,i}^*$, and satisfies $\Delta w_{q_{0,i}^* q_{0,i}^*}(t-1)\Delta d_{q_{0,i}^*}(t) - \Delta w_{qq}(t-1)\Delta d_q(t) \geq \epsilon > 0$ for $t \geq 1$, $\forall q \in \Lambda_{t,i} \setminus \{q_{0,i}^*\}$, i.e., the $q_{0,i}^*{}^{th}$ robot's change in its social weight shows the most positively aligned or the least negatively aligned change in its detection confidence score compared to other robots including the $i^{th}$ robot and its neighbors for time $t \geq 1$. This implies $\sum_{s=1}^{t} \Delta w_{q_{0,i}^* q_{0,i}^*}(s-1)\Delta d_{q_{0,i}^*}(s) - \sum_{s=1}^{t} \Delta w_{qq}(s-1)\Delta d_q(s) \geq \epsilon t$, which is a special case under assumption 5 when $\beta = 1$. Considering equation (12), this further implies that the estimate $\hat{\mathbf{x}}_{t,B}^{I_{q_{0,i}^*}}$ is more accurately estimating the target position $\mathbf{x}_{t,B}$ compared to other estimates in the weighted fusion, for $t \geq 1$. In practice, assumption 5 would be satisfied for any such scenario where the best ($q_{t,i}^*{}^{th}$) robot stays fixed for some finite time duration and may change intermittently over time. In the simulation studies (section 5), such intermittent changes result from temporary/permanent sensor failures, intermittent communication link loss, and target visual loss.

**Theorem 3.** *Under assumptions 4 and 5, given $e_{w2}t^{1-\beta} < \epsilon e_{w1}$ (check equation (19)), $\forall i \in [N]$, AOL-2P algorithm's weights $w_{ij}(t)$ satisfy the following:*

$$\lim_{t\to\infty} w_{ij}(t) = 0, \quad \forall j \in \Lambda_{\infty,i} \setminus \{q_{\infty,i}^*\} \tag{34}$$

*and*

$$\lim_{t\to\infty} w_{iq_{\infty,i}^*}(t) = 1 \tag{35}$$

*where $0 < \epsilon \ll 1$, $\beta \in (0,1]$, and $q_{\infty,i}^*$ is the index of the robot that exhibits the greatest cumulative sum of the difference-based correction term among the robots $j \in \Lambda_{\infty,i}$.*

*Remark 7:* Since $0 < \epsilon \ll 1$, the condition $e_{w2}t^{1-\beta} < \epsilon e_{w1}$ further yields $e_{w2}t^{1-\beta} \ll e_{w1}$. This can be satisfied either by choosing $e_{w2} = 0$, or by choosing a time-varying $e_{w2}$ such that $e_{w2}(t) \propto t^{-c}$ for $c > 0$, where $1 - \beta - c \leq 0$. In practice, since the AOL-2P algorithm involves a periodic reset with a period of $T_p$ discrete-time steps, this condition implies $e_{w2}T_p{}^{1-\beta} \ll e_{w1}$. This condition can be satisfied when $e_{w2}$ is chosen to be much smaller than $e_{w1}$ or $e_{w2} \approx 0$.

*Proof.* Consider $w_{ij}(t)$ given by equation (13); given $\hat{w}_{ii}(0) = 1$, $\forall i \in [N]$, use the update rule given by equations (19) and (20) to get

$$w_{ij}(t) = \mathbb{1}(j \in \Lambda_{t,i}) \frac{\hat{w}_{jj}(t)}{\sum_{q\in\Lambda_{t,i}} \hat{w}_{qq}(t)} = \mathbb{1}(j \in \Lambda_{t,i}) \frac{\exp(R_{t,j}^w)}{\sum_{q\in\Lambda_{t,i}} \exp(R_{t,q}^w)} \tag{36}$$

where $R_{t,i}^w = \sum_{s=1}^t r_{t,i}^w$ and $\mathbb{1}(\cdot)$ is the indicator function; $\mathbb{1}(j \in \Lambda_{t,i}) = 1$ if the condition $j \in \Lambda_{t,i}$ is satisfied, otherwise $\mathbb{1}(j \in \Lambda_i^{k,l}) = 0$. Dividing both numerator and denominator in equation (36) by $\exp(R_{t,q_{t,i}^*}^w)$, we get

$$w_{ij}(t) = \mathbb{1}(j \in \Lambda_{t,i}) \frac{\exp(R_{t,j}^w - R_{t,q_{t,i}^*}^w)}{\sum_{q \in \Lambda_{t,i}} \exp(R_{t,q}^w - R_{t,q_{t,i}^*}^w)} = \mathbb{1}(j \in \Lambda_{t,i}) \frac{\exp(R_{t,j}^w - R_{t,q_{t,i}^*}^w)}{1 + \sum_{q \in \Lambda_{t,i} \setminus \{q_{t,i}^*\}} \exp(R_{t,q}^w - R_{t,q_{t,i}^*}^w)} \tag{37}$$

Using the reward definition of $r_{t,i}^w$ as given by equation (19), since the perturbations occur just after $t = 0$, therefore at $t = 1$, $e_p^{w_i} = Unif.(0, p_{mag})$ and for $t > 1$, $e_p^{w_i} = 0$ (assumed for ease in analysis, without the loss of generality), $\forall i \in [N]$, $R_{t,i}^w$ can be written as

$$R_{t,i}^w = e_p^{w_i}(1 - d_{1,i}) + e_{w1} \sum_{s=1}^t \Delta w_{ii}(s-1)\Delta d_i(s) + e_{w2} \sum_{s=1}^t w_{ii}(s-1)d_{s,i} \tag{38}$$

where $\Delta w_{ii}(t-1) = w_{ii}(t-1) - w_{ii}(t-2)$ and $\Delta d_i(t) = d_{t,i} - d_{t-1,i}$. Using equation (38), $\forall q \in \Lambda_{t,i}$, we get

$$\begin{aligned} R_{t,q}^w - R_{t,q_{t,i}^*}^w = \;& e_p^{w_q}(1 - d_{1,q}) - e_p^{w_{q_{t,i}^*}}(1 - d_{1,q_{t,i}^*}) \\ &+ e_{w1}\left(\sum_{s=1}^t \Delta w_{qq}(s-1)\Delta d_q(s) - \sum_{s=1}^t \Delta w_{q_{t,i}^* q_{t,i}^*}(s-1)\Delta d_{q_{t,i}^*}(s)\right) \\ &+ e_{w2}\sum_{s=1}^t \left(w_{qq}(s-1)d_{s,q} - w_{q_{t,i}^* q_{t,i}^*}(s-1)d_{s,q_{t,i}^*}\right) \end{aligned} \tag{39}$$

Since $w_{qq}(t) \in [0,1]$ and $d_{t,q} \in [0,1]$, note that $\Delta w_{qq}(t-1)\Delta d_q(t) \in [-1,1]$, $\forall q \in \Lambda_{t,i}$. Using this along with assumption 5, we get

$$0 < \epsilon t^\beta \leq \sum_{s=1}^t \Delta w_{q_{t,i}^* q_{t,i}^*}(s-1)\Delta d_{q_{t,i}^*}(s) - \sum_{s=1}^t \Delta w_{qq}(s-1)\Delta d_q(s) \leq t \tag{40}$$

where $0 < \epsilon \ll 1$ and $\beta \in (0,1]$. Further, note that $w_{qq}(t-1)d_{t,q} \in [0,1]$. Therefore,

$$-t \leq \sum_{s=1}^t \left(w_{qq}(s-1)d_{s,q} - w_{q_{t,i}^* q_{t,i}^*}(s-1)d_{s,q_{t,i}^*}\right) \leq t \tag{41}$$

Using equations (40) and (41) in equation (39), $\forall q \in \Lambda_{\infty,i} \setminus \{q_{\infty,i}^*\}$, we get

$$\begin{aligned} e_p^{w_q}(1 - d_{1,q}) - e_p^{w_{q_{\infty,i}^*}}(1 - d_{1,q_{\infty,i}^*}) - e_{w1}t - e_{w2}t \\ \leq (R_{t,q}^w - R_{t,q_{t,i}^*}^w) \leq \\ e_p^{w_q}(1 - d_{1,q}) - e_p^{w_{q_{\infty,i}^*}}(1 - d_{1,q_{\infty,i}^*}) - e_{w1}\epsilon t^\beta + e_{w2}t \end{aligned} \tag{42}$$

Given $e_{w2}t^{1-\beta} < \epsilon e_{w1}$, taking $\lim_{t \to \infty}(\cdot)$ on equation (42), $\forall q \in \Lambda_{\infty,i} \setminus \{q_{\infty,i}^*\}$, implies

$$(R_{t,q}^w - R_{t,q_{t,i}^*}^w) \to -\infty \quad \text{as} \quad t \to \infty \tag{43}$$

Taking $\lim_{t \to \infty}(\cdot)$ on both sides of equation (37) under assumption 4, and using equation (43), $\forall j \in \Lambda_{\infty,i} \setminus \{q_{\infty,i}^*\}$, we get

$$\lim_{t \to \infty} w_{ij}(t) = 0 \tag{44}$$

and for $j = q_{\infty,i}^*$, we get

$$\lim_{t \to \infty} w_{iq_{t,i}^*}(t) = 1 \tag{45}$$

Equations (44) and (45) complete the proof. $\qquad\square$

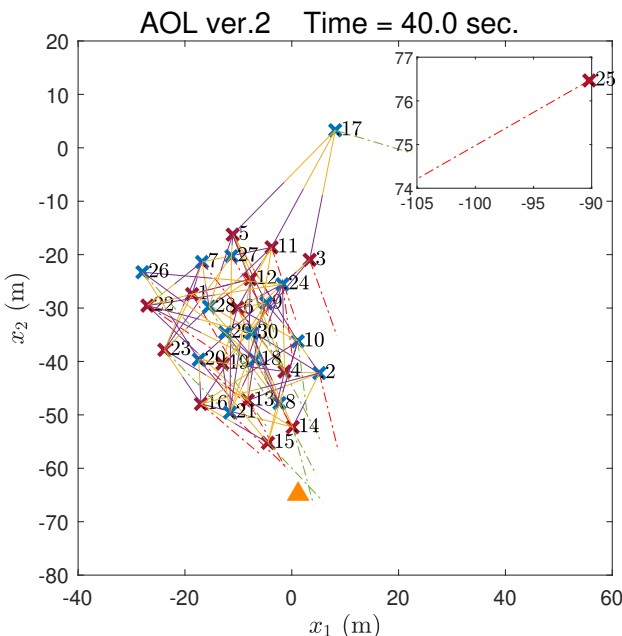

Figure 4: Screenshot of the AOL-1P simulation; orange triangle – target, blue x – robots with functional proprioception, red x – robots with faulty proprioception, green dash-dot lines – functional exteroception, red dash-dot lines – faulty exteroception, purple-yellow links – communication links, the flow of information is from purple towards yellow.

## 5  Performance Evaluation

The proposed AOL framework is evaluated using a simulation setup involving $N = 5, 10, 20, 30$ robots executing the cooperative target monitoring task discussed in the problem formulation. The communication range $R_{comm.}$ and the communication link drop probability $p_{ld}$ are set to be 30 $m$ and 0.1, respectively, with the limit on the number of communication neighbors as $n_l = 3$. The exteroceptive sensor model's parameters are set as $R_{FOV} = 15$ $m$, $\theta_{FOV} = 160$ *degrees*, with the target visual loss probability as $p_{vl} = 0.1$. The parameters for the detection confidence model are set as $r_o = 10$ $m$, and $b_o = 0.1$.

The simulation results are averaged over 100 simulation runs. Each run involves a time horizon of $T = 600$ discrete time steps, with a sampling period of $\Delta T = 0.1$ *sec.* The robots follow the control law described in the problem formulation while trying to maintain a safe distance of 8 $m$ from the target (more details in the supplementary document). The target randomly changes its velocity and yaw rate after every 5 *seconds*. The robots and the target always stay inside a square region of side length 100 $m$ by overriding their control laws to get away from the region boundary. At the start of each simulation run, the robots are always spawned near the center of the square region, whereas the target is spawned randomly but sufficiently near to the robots so that at least one of the robots is likely to detect it at the start of the run. This is done since the main focus of this paper is not the target search but target detection and monitoring.

In the considered adverse scenario for the simulation, 50% of the total robots are chosen randomly at times 0, 10, 20, 30, and 50 *seconds*, that exhibit temporary failures in their exteroceptive sensors. Further, 50% of the total robots are chosen randomly at time 10 *sec.*, that exhibit permanent failures in their proprioceptive sensors. The noise $\mu_{t,i}^x$ and $\mu_{t,i}^\phi$ in the proprioceptive sensors (equations (4)) that are functional is assumed to be Gaussian with a mean of $0.01m$ and 0.02 deg., respectively, with a covariance of $0.01m^2$ and $0.01rad.^2$, respectively. The proprioceptive sensors that fail have noise terms that are still Gaussian but with large bias (mean) of $5m$ and 10 deg., respectively, with either a covariance of $0.01m^2$ or $5m^2$ with equal probability and $0.01rad.^2$ or $5rad.^2$ with equal probability, respectively, since the bias and the covariance may be uncorrelated. Similarly, the noise $\nu_{t,i}$ in the exteroceptive sensors (equation (6)) that are functional is

| Algorithm | Features | Remarks |
|---|---|---|
| KCF | Not adaptive; covariance-based | performs poorly when sensor failures do not lead to an increase in estimation covariance |
| ACF | Not adaptive; equal weights | performs poorly under adverse scenarios - considerable no. of robots undergoing failures |
| AOL-C | adaptive; comparative loss $\rightarrow$ local weights; detection confidence as a reward $\rightarrow$ social weights | $2^{nd}$ best AOL variant; performs significantly better than AOL-2P, KCF, ACF in adverse scenarios |
| AOL-1P | adaptive; perturbation-greedy reward $\rightarrow$ local weights; detection confidence as a reward $\rightarrow$ social weights | the best AOL variant; performs significantly better than AOL-2P, KCF, ACF in adverse scenarios |
| AOL-2P | adaptive; perturbation-greedy reward $\rightarrow$ local weights; perturbation-greedy reward $\rightarrow$ social weights | the worst AOL variant; performs better than KCF, ACF in adverse scenarios |

Table 1: Comparison of the three AOL variants with Kalman-Consensus Fusion (KCF) and Averaging-Consensus Fusion (ACF)

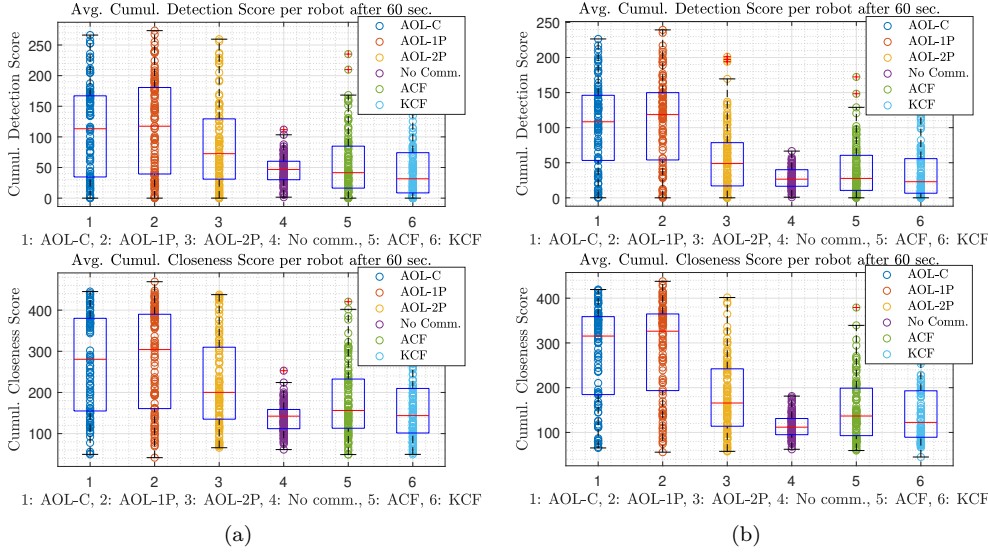

Figure 5: (a) Comparison results; 100 sim. runs for each method, $N = 5$. (b) Comparison results; 100 sim. runs for each method, $N = 10$.

assumed to be Gaussian with a mean of $0.01m$ with a covariance of $0.01m^2$. For the exteroceptive sensors that fail, the noise term is still Gaussian but with a large bias (mean) of $10m$ with a covariance of $0.01m^2$ or $5m^2$ with equal probability since the fault may or may not lead to an increase in the covariance.

The algorithms are evaluated based on two types of scores – 1) cumulative average detection score per robot: $\frac{1}{N} \sum_{i=1}^{N} \sum_{t=1}^{T} d_{t,i}$, and 2) cumulative average closeness score per robot: $\frac{1}{N} \sum_{i=1}^{N} \sum_{t=1}^{T} l_{t,i}^c$, where $l_{t,i}^c := \min \left\{ \frac{8}{||\mathbf{x}_{t,i} - \mathbf{x}_{t,B}||}, 1 \right\}$. Note that the robots try to maintain a distance of $8$ $m$ from the target. The cumulative average detection score per robot is a measure of the target detection performance of the robots

in the swarm; a higher score means that a larger number of robots are able to detect and monitor the target for a longer duration. As the number of robots in the swarm increases, crowding occurs, making it harder for all the robots to be able to detect and monitor the target – the robots closest to the target are likely to block the robots behind them in the swarm. Therefore, in such scenarios, the cumulative average closeness score per robot becomes a better criterion to judge the swarm's performance, since it is a measure of how close the robots are to the target. Note that in scenarios where the target can be fully observable to the swarm (i.e., all the robots can view the target simultaneously), the cumulative average detection score and the cumulative average closeness score play an equivalent role in judging the swarm's performance. Whereas their roles differ in scenarios where the target is partially observable to the swarm (i.e., not all robots can view the target simultaneously).

For all three AOL variants, a simulation-based parametric study is carried out to find a suitable set of parameters that result in desirable performance. Based on the parametric study, the parameters for AOL-C are set as $D_o = 15\ m$, $T_p = 15$, $\eta_w = 15$, and $\eta_\alpha = 0.01$; that of AOL-1P are set as $T_p = 15$, $\eta_w = 15$, $e_{a1} = 10$, $e_{a2} = 0.1$, and $p_{mag} = 0.1$; that of AOL-2P are set as $T_p = 15$, $e_{w1} = 1$, $e_{w2} = 0.01$, $e_{a1} = 20$, $e_{a2} = 5$, and $p_{mag} = 0.1$. Further, the three variants of AOL are compared with two baselines (check Table 1) – Average-Consensus Fusion (ACF; equations (9), (10) and (12) with equal weights), and Kalman-Consensus Fusion (KCF) – more details in the supplementary document.

A snapshot of the simulation of the AOL-1P in adverse conditions is shown in Fig. 4. Simulation videos are submitted as a supplementary file.

Figures 5a, 5b, 6a, and 6b show the comparison results for $N = 5$, $N = 10$, $N = 20$, and $N = 30$, respectively. For each method, 100 simulations are run, and the resulting cumulative scores are collected as points, which are then used to generate their corresponding box plots. Note that on each box, the central mark indicates the median, and the bottom and top edges of the box indicate the $25^{th}$ and $75^{th}$ percentiles, respectively. For $N = 5$, $N = 10$, $N = 20$, and $N = 30$, the *best* variant AOL-1P performs 182.2%, 329.7%, 463%, and 652% better in terms of cumulative detection score and 94.7%, 138.6%, 167.3%, and 150.4% better in terms of cumulative closeness score than the *best* baseline ACF, respectively. Further, for $N = 5$, $N = 10$, $N = 20$, and $N = 30$, AOL-1P performs 3.62%, 9.41%, 8.86%, and 1.1% better in terms of cumulative detection score and 8.37%, 3.42%, 4.5%, and 0.0% better in terms of cumulative closeness score than the *second best* AOL variant, AOL-C. Among the baselines, ACF performs better than KCF for all the cases in terms of both types of scores.

The three AOL variants perform significantly better compared to KCF; since KCF is a covariance-based method, it is unable to handle adverse conditions involving temporary or permanent sensor failures inducing large biases that may or may not increase the covariance. Since the weights are adapted through an online learning process in the AOL variants, their performance is better than ACF as well; unlike the AOL variants, ACF gives equal weight to all the input estimates being fused in the weighted estimate fusion process. The no-communication case performs better than KCF and comparably to ACF; unlike the AOL variants, KCF and ACF do not have a mechanism to adaptively filter out the corrupt information coming from the robots undergoing sensor failures, due to which the corrupt information propagates throughout the communication network without any inhibition, thereby corrupting even those robots having functional sensors.

Using the detection confidence $d_{t,i}$ as a reward signal does better than having a *perturbation-greedy* reward-based learning strategy for the update of weights $\hat{w}_{ii}(t)$ in the social learning layer, which is quite evident by the improved performance of AOL-1P and AOL-C compared to the AOL-2P. However, using a *perturbation-greedy* reward-based learning update strategy for the update of weights $\hat{\alpha}_i(t)$ and $\hat{\alpha}'_i(t)$ in the local learning layer, as in AOL-1P, does better than using a *comparative* Euclidean distance based loss function as in AOL-C. Moreover, for AOL-1P, when $N$ is increased from 5 to 10, the cumulative detection score and the closeness score increase by 1.02% and 7.3%, which then decrease by 20.2% and 11.0% from $N = 10$ to $N = 20$, and then decrease further by 22.4% and 18.8% from $N = 20$ to $N = 30$, respectively. Similar trends are observed for other algorithms as well. These observations are justified since the robots, while trying to avoid collisions, effectively push each other, therefore, making it harder for all the robots to be able to detect and stay close to the target when crowding occurs as the number of robots $N$ is further increased to much larger values.

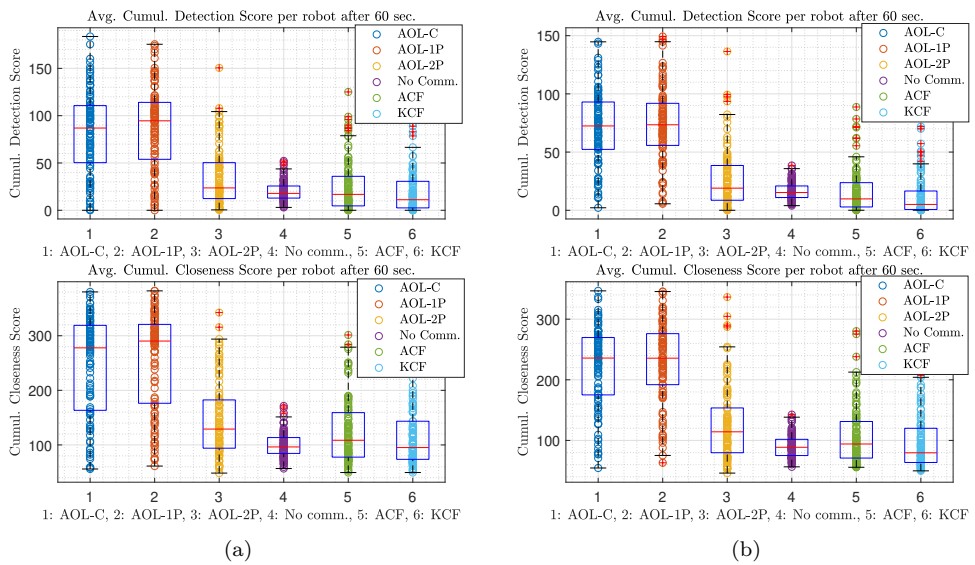

Figure 6: (a) Comparison results; 100 sim. runs for each method, $N = 20$. (b) Comparison results; 100 sim. runs for each method, $N = 30$.

Further, Sim2Real aspects of the top two performing AOL variants, AOL-1P and AOL-C, are validated using a ROS-Gazebo setup (simulation video submitted as a supplementary file; check the supplementary document).

## 6    Conclusion

This paper proposes a novel Autonomous Online Learning (AOL) framework for the decentralized monitoring of an agile target using a swarm of robots undergoing sensor failures, communication link drops, and target visual loss and operating without the assistance of a supervisor or a landmark, i.e., without the availability of ground truth regarding pose information. In the AOL framework, a decentralized online learning mechanism driven by reward-like signals (based on detection confidence) is intertwined with an implicit adaptive consensus-based, two-layered, weighted information fusion process, thereby allowing the robotic swarm to exhibit improved robustness and adaptability. Within the AOL framework, three variants are proposed in order to study the effect of using different loss or reward function designs in the learning phase. A novel *perturbation-greedy* reward design is introduced in the learning process of two AOL variants, leading to exploration-exploitation in their information fusion's weights' space. For the three AOL variants, convergence analysis of the weights involved in their weighted information fusion process shows that the weights converge under reasonable assumptions. Moreover, the AOL algorithms involve analytic expressions making them computationally inexpensive and therefore, ideal for use in robotic swarms. Simulation results show that among the three variants, AOL-1P performs the best in terms of detection and closeness scores, owing to its use of the perturbation-greedy reward for learning the weights that belong to the local estimation phase, and detection confidence-based reward for learning the weights that belong to the social estimation phase. AOL-1P performs 182.2% to 652% and 94.7% to 150.4% better than the baselines in terms of cumulative average detection score per robot and cumulative average closeness score per robot, respectively, as the total number of robots is increased from 5 to 30. The Sim2Real aspects of the top two performing AOL variants are evaluated using a ROS-Gazebo setup. The current variants of AOL are synchronous; the asynchronous variant is left as future work.

**Broader Impact Statement**

The reward-based autonomous online learning framework proposed in this paper, called the AOL framework, is an adaptive information fusion framework that can be applied to various reward-based multi-agent applications, where the agents are required to be smart about which neighbors should be trusted when for a particular type of information, to increase their rewards as well as help others in the communication network. The problem considered in this paper, i.e., target search, tracking, and monitoring, can be used for many applications, like search and rescue, firefighting, disaster relief, convoy protection, wildlife monitoring, surveillance, etc. Moreover, the AOL framework can be useful in the search and detection of societal threats on the web by the use of multiple collaborative bots – the weights involved in the information fusion indicate which neighboring bot is detecting an increased threat level, thus helping in pinpointing the source of the activity in the overall network. Further, AOL can also be used in e-commerce applications, such as trading bots collaborating to increase profits, recommendation bots collaborating to improve customer reviews, etc.

**Acknowledgments**

The authors would like to thank the Kotak IISc AI-ML Centre (KIAC) for their support.

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

# A    Appendix

## A.1    Nomenclature

| | |
|---|---|
| $\lvert \cdot \rvert$ | Cardinality operator (cardinality of a set) |
| $\lVert \cdot \rVert$ | 2-norm or Euclidean norm |
| $N$ | Total number of robots in the swarm |
| $\Delta T$ | Sampling period (seconds) |
| $\mathbf{x}_{t,i}$ | The $i^{th}$ robot's 2-D position vector (in $m$) at time $t$ |
| $\bar{\mathbf{v}}_{t,i}$ | The $i^{th}$ robot's body-axis velocity vector ($m/s$) at time $t$ |
| $\phi_{t,i}$ | The $i^{th}$ robot's heading angle (radians) at time $t$ |
| $\bar{w}_{t,i}$ | The $i^{th}$ robot's yaw rate ($rad/s$) at time $t$ |
| $\mathbf{x}_{t,B}$ | The target's 2-D position vector (in $m$) at time $t$ |
| $\bar{\mathbf{v}}_{t,B}$ | The target's body-axis velocity vector ($m/s$) at time $t$ |
| $\phi_{t,B}$ | The target's heading angle (radians) at time $t$ |
| $\bar{w}_{t,B}$ | The target's yaw rate ($rad/s$) at time $t$ |
| $R_{comm.}$ | The range of communication |
| $p_{ld}$ | The communication link drop probability |
| $G_t$ | A uni-directional dynamic graph representing communication network topology among the robots |
| $\mathbf{A}_t$ | Adjacency matrix of the graph $G_t$ |
| $U_{ij}(0,1)$ | A uniform random variable, $U_{ij}(0,1) \in [0,1]$, for $i,j \in [N]$ |
| $\Omega_{t,i}$ | The neighbor set of the $i^{th}$ robot as per the graph $G_t$, at time $t$ |
| $n_{t,i}$ | The number of communicating neighbors of the $i^{th}$ robot at time $t$, $n_{t,i} := \lvert \Omega_{t,i} \rvert$ |
| $n_l$ | Limit on the number of neighbors the $i^{th}$ robot can have, i.e., $n_{t,i} \leq n_l$, where $n_l \in \{1, 2, \cdots, N\}$ |
| $\hat{\mathbf{x}}_{t,i}^{P_i}$ | The $i^{th}$ robot's proprioceptive sensor suite's estimate of its 2-D position |
| $\hat{\phi}_{t,i}^{P_i}$ | The $i^{th}$ robot's proprioceptive sensor suite's estimate of its yaw angle |
| $\mu_{t,i}^{x}$ | Bounded arbitrary noise in the $i^{th}$ robot's proprioceptive sensor suite's estimate of its 2-D position |
| $\mu_{t,i}^{\phi}$ | Bounded arbitrary noise in the $i^{th}$ robot's proprioceptive sensor suite's estimate of its yaw angle |

| | |
|---|---|
| $d_{t,i}$ | The $i^{th}$ robot's exteroceptive sensor suite's target detection confidence, $d_{t,i} \in [0,1]$ |
| $\Delta \hat{\mathbf{x}}_{t,B}^{E_i}$ | The $i^{th}$ robot's exteroceptive sensor suite's estimate of the target's relative position |
| $R_{FOV}$ | Detection range of the exteroceptive sensor suite |
| $\theta_{FOV}$ | Angle-of-view of the exteroceptive sensor suite |
| $p_{vl}$ | The probability of target visual loss |
| $\nu_{t,i}$ | Bounded arbitrary noise in the $i^{th}$ robot's exteroception's estimate of the target's relative position |
| $\hat{\mathbf{x}}_{t,B}^{S_i}$ | The combined sensor estimate of the target's position; combines the estimates from proprioception and exteroception |
| $\bar{\mathbf{v}}_{t,i}^{R}$ | The $i^{th}$ robot's velocity reference command signal |
| $\Delta \bar{\mathbf{v}}_{t,i}$ | The $i^{th}$ robot's velocity correction control signal |
| $T$ | Discrete-time horizon |
| $\hat{\mathbf{x}}_{t,B}^{I_i}$ | The $i^{th}$ robot's intermediate estimate of the target's position |
| $\hat{\mathbf{x}}_{t,B}^{i}$ | The $i^{th}$ robot's final estimate of the target's position |
| $\alpha_i(t)$ | Weights involved in the local estimation phase |
| $\Lambda_{t,i}$ | $\Lambda_{t,i} = \Omega_{t,i} \cup \{i\}$ |
| $w_{ij}(t)$ | Weights involved in the social estimation phase |
| $\hat{\alpha}_i(t), \hat{\alpha}_i'(t)$ | Weights updated in the local learning layer |
| $\hat{w}_{ii}(t)$ | Weights updated in the social learning layer |
| $l_{t,i}^{w}$ | Loss function involved in weights update in the social learning layer of AOL-C and AOL-1P, $l_{t,i}^{w} := (1 - d_{t,i})$ |
| $L_{t,i}^{w}$ | $L_{t,i}^{w} := \sum_{s=1}^{t} l_{s,i}^{w}$ |
| $\eta_w$ | Learning rate parameter for the social learning layer in AOL-C and AOL-1P |
| $\eta_\alpha$ | Learning rate parameter for the local learning layer in AOL-C |
| $r_{t,i}^{\alpha}, r_{t,i}^{\alpha'}$ | *Perturbation-greedy* reward functions involved in weights update in the local learning layer of AOL-1P and AOL-2P |
| $R_{t,i}^{\alpha}, R_{t,i}^{\alpha'}$ | $R_{t,i}^{\alpha} := \sum_{s=1}^{t} r_{s,i}^{\alpha}$ and $R_{t,i}^{\alpha'} := \sum_{s=1}^{t} r_{s,i}^{\alpha'}$ |

| | |
|---|---|
| $e_{a1}$ | Learning rate parameter associated with the difference-based correction term in $r^\alpha_{t,i}$ and $r^{\alpha'}_{t,i}$ |
| $e_{a2}$ | Learning rate parameter associated with the inertia term in $r^\alpha_{t,i}$ and $r^{\alpha'}_{t,i}$ |
| $e_p, e'_p$ | Perturbation signals associated with the perturbation terms in $r^\alpha_{t,i}$ and $r^{\alpha'}_{t,i}$ |
| $\Delta\alpha_i(t-1)$ | $\Delta\alpha_i(t-1) := \alpha_i(t-1) - \alpha_i(t-2)$ |
| $\Delta d_i(t)$ | $\Delta d_i(t) := d_{t,i} - d_{t-1,i}$ |
| $l^\alpha_{t,i}, l^{\alpha'}_{t,i}$ | $l^\alpha_{t,i} := \min\{||\hat{\mathbf{x}}^{S_i}_{t,B} - \hat{\mathbf{x}}^{I_{r^*_{t,i}}}_{t,B}||/D_o, 1\}, \ l^{\alpha'}_{t,i} := \min\{||\hat{\mathbf{x}}^i_{t-1,B} - \hat{\mathbf{x}}^{I_{r^*_{t,i}}}_{t,B}||/D_o, 1\}$ |
| $D_o$ | A normalization parameter |
| $r^*_{t,i}$ | $r^*_{t,i} := \arg\max_{j\in\Lambda_{t,i}} w_{ij}(t-1)$ |
| $r^w_{t,i}$ | *Perturbation-greedy* reward functions involved in weights update in the social learning layer of AOL-2P |
| $R^w_{t,i}$ | $R^w_{t,i} := \sum_{s=1}^t r^w_{s,i}$ |
| $e_{w1}$ | Learning rate parameter associated with the difference-based correction term in $r^w_{t,i}$ |
| $e_{w2}$ | Learning rate parameter associated with the inertia term in $r^w_{t,i}$ |
| $e^{w_i}_p$ | Perturbation signal associated with the perturbation term in $r^w_{t,i}$ |
| $\Delta w_{ii}(t-1)$ | $\Delta w_{ii}(t-1) := w_{ii}(t-1) - w_{ii}(t-2)$ |
| $j^*_{t,i}$ | $j^*_{t,i} := \arg\min_{j\in\Lambda_{t,i}} L^w_{t,j}$ |
| $\Lambda_{\infty,i}$ | $\lim_{t\to\infty} \Lambda_{t,i} = \Lambda_{\infty,i}$ |
| $j^*_{\infty,i}$ | $\lim_{t\to\infty} j^*_{t,i} = j^*_{\infty,i}$ |
| $q^*_{t,i}$ | $q^*_{t,i} := \arg\max_{j\in\Lambda_{t,i}} \sum_{s=1}^t \Delta w_{jj}(s-1)\Delta d_j(s)$ |
| $q^*_{\infty,i}$ | $\lim_{t\to\infty} q^*_{t,i} = q^*_{\infty,i}$ |

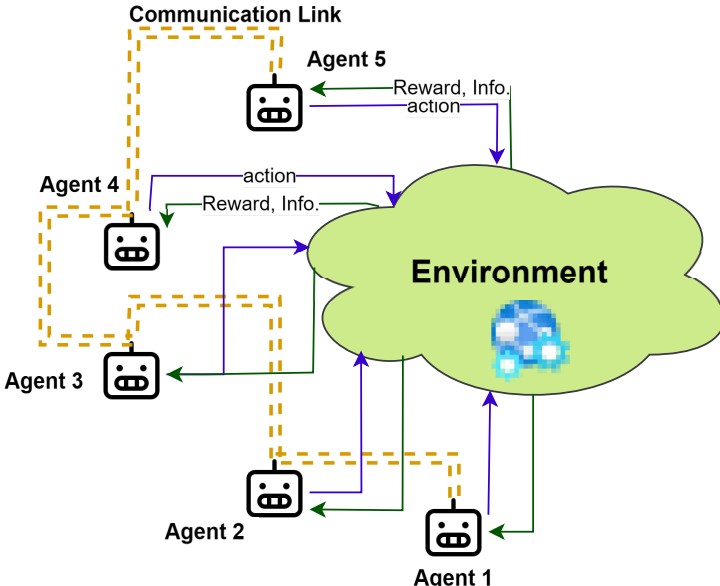

Figure 7: The AOL framework can be useful in a multi-agent problem setting where the agents have a shared environment, both a common objective and an information space, and can fuse information from their communicating neighbors to increase their reward.

## A.2 Generalizability of AOL framework

The idea behind the proposed AOL framework can be implemented in any such multi-agent problem setting or application where:

- The agents have a shared environment, a common objective, and a common information space (estimation/prediction space).

- An agent's performance is governed by a reward-like signal.

- How the agents fuse information from their neighbors (i.e., which neighbor to listen to) influences their performance or reward – accurate or quality information leads to better reward.

The proposed AOL framework can be viewed as a reward-driven, online learning-based, adaptive information fusion framework that can be used by agents in a multi-agent problem setting (Fig. 7) to improve their reward/performance. Therefore, the AOL framework is useful in problems where the reward/performance of each agent is dependent on the quality/accuracy of the information that the agent has – the dependence can be direct (the fused information is directly responsible for reward generation) or indirect (the fused information is used to make a decision which is then responsible for reward generation).

The problem of cooperative target monitoring considered in this paper involves an indirect dependence of the reward (target detection confidence) on the fused information (target position estimates). The reward-like signal, called target detection confidence or score ($d_{t,i}$), is produced by each robot's exteroceptive sensor suite. The communicating robots utilize the AOL framework to determine what information should be given more weight in the weighted information fusion that improves their cumulative target detection score (cumulative reward). Therefore, the AOL framework (along with its theoretical analysis) works for any particular set of tasks, robots, or robot dynamics as long as the above-mentioned general conditions are satisfied. Examples include monitoring and controlling the spread of fire (target) in a forest, wildlife monitoring, search and rescue, etc.

Similar problems where the AOL framework can be used are searching and detecting societal threats on the web by using multiple collaborative bots, trading bots collaborating to increase profits, recommendation bots collaborating to improve customer reviews, etc.

## A.3  Pseudo-code

---

**Algorithm 1** AOL-C (for the $i^{th}$ robot, $i \in [N]$)

---

    **Choose:** $D_o > 0$, $\eta_\alpha > 0$, $\eta_w > 0$, $T_p \geq 1$ (integer-valued)
    **Initialize:** $\hat{\alpha}_i(0) = \hat{\alpha}'_i(0) = 1$, $w_{ii}(0) = 1$, $\hat{\mathbf{x}}^i_{0,B} = \hat{\mathbf{x}}^{S_i}_{0,B}$
    **Input:** $t$, $d_{t,i}$, $\hat{\alpha}_i(t-1)$, $\hat{\alpha}'_i(t-1)$, $\hat{w}_{ii}(t-1)$, $\hat{\mathbf{x}}^{S_i}_{t,B}$, $\hat{\mathbf{x}}^i_{t-1,B}$
    **Output:** $\hat{\alpha}_i(t)$, $\hat{\alpha}'_i(t)$, $\hat{w}_{ii}(t)$, $\hat{\mathbf{x}}^i_{t,B}$

1:   $\alpha_i(t-1) = \frac{\hat{\alpha}_i(t-1)}{\hat{\alpha}_i(t-1) + \hat{\alpha}'_i(t-1)}$
2:   **if** $d_{t,i} > 0$ **then**
3:       $\hat{\mathbf{x}}^{I_i}_{t,B} = \alpha_i(t-1)\hat{\mathbf{x}}^{S_i}_{t,B} + (1 - \alpha_i(t-1))\hat{\mathbf{x}}^i_{t-1,B}$
4:   **else**
5:       $\hat{\mathbf{x}}^{I_i}_{t,B} = \hat{\mathbf{x}}^i_{t-1,B}$
6:   **end if**
7:   Broadcast $\{t, i, d_{t,i}, \hat{\mathbf{x}}^{I_i}_{t,B}, \hat{\mathbf{x}}^i_{t-1,B}, \hat{w}_{ii}(t-1)\}$ and receive $\{t, j, d_{t,j}, \hat{\mathbf{x}}^{I_j}_{t,B}, \hat{\mathbf{x}}^j_{t-1,B}, \hat{w}_{jj}(t-1)\}$ from the communicating neighboring robots $j \in \Omega_{t,i}$
8:   $w_{ij}(t-1) = \frac{\hat{w}_{jj}(t-1)}{\sum_{\forall j' \in \Lambda_{t,i}} \hat{w}_{j'j'}(t-1)}$
9:   $\hat{\mathbf{x}}^i_{t,B} = \sum_{\forall j \in \Lambda_{t,i}} w_{ij}(t-1)\hat{\mathbf{x}}^{I_j}_{t,B}$, where $\Lambda_{t,i} = \Omega_{t,i} \cup \{i\}$
10:   $r^*_{t,i} := \arg\max_{j \in \Lambda_{t,i}} w_{ij}(t-1)$
11:   $l^\alpha_{t,i} = \min\{\|\hat{\mathbf{x}}^{S_i}_{t,B} - \hat{\mathbf{x}}^{I_{r^*_{t,i}}}_{t,B}\|/D_o, 1\}$;   $l^{\alpha'}_{t,i} = \min\{\|\hat{\mathbf{x}}^i_{t-1,B} - \hat{\mathbf{x}}^{I_{r^*_{t,i}}}_{t,B}\|/D_o, 1\}$
12:   *Periodic reset*: set the weights $\hat{\alpha}_i(t-1)$, $\hat{\alpha}'_i(t-1)$, and $\hat{w}_{ii}(t-1)$ to 1 after every $T_p$ discrete time steps
13:   $\hat{\alpha}_i(t) = \hat{\alpha}_i(t-1)\exp(-\eta_\alpha l^\alpha_{t,i})$;   $\hat{\alpha}'_i(t) = \hat{\alpha}'_i(t-1)\exp(-\eta_\alpha l^{\alpha'}_{t,i})$
14:   $\hat{w}_{ii}(t) = \hat{w}_{ii}(t-1)\exp(-\eta_w(1 - d_{t,i}))$

---

---

**Algorithm 2** AOL-1P (for the $i^{th}$ robot, $i \in [N]$)

---

**Choose:** $D_o > 0$, $\eta_\alpha > 0$, $\eta_w > 0$, $T_p \geq 1$ (integer-valued), $e_{a1} > 0$, $e_{a2} \geq 0$, $p_{mag} > 0$

**Initialize:** $\hat{\alpha}_i(0) = \hat{\alpha}'_i(0) = \hat{\alpha}_i(-1) = \hat{\alpha}'_i(-1) = 1$, $w_{ii}(0) = 1$, $\hat{\mathbf{x}}^i_{0,B} = \hat{\mathbf{x}}^{S_i}_{0,B}$

**Input:** $t$, $d_{t,i}$, $\hat{\alpha}_i(t-1)$, $\hat{\alpha}'_i(t-1)$, $\hat{w}_{ii}(t-1)$, $\hat{\mathbf{x}}^{S_i}_{t,B}$, $\hat{\mathbf{x}}^i_{t-1,B}$, $d_{t-1,i}$, $\alpha_i(t-2)$

**Output:** $\hat{\alpha}_i(t)$, $\hat{\alpha}'_i(t)$, $\hat{w}_{ii}(t)$, $\hat{\mathbf{x}}^i_{t,B}$

1: $\alpha_i(t-1) = \frac{\hat{\alpha}_i(t-1)}{\hat{\alpha}_i(t-1) + \hat{\alpha}'_i(t-1)}$

2: **if** $d_{t,i} > 0$ **then**

3:     $\hat{\mathbf{x}}^{I_i}_{t,B} = \alpha_i(t-1)\hat{\mathbf{x}}^{S_i}_{t,B} + (1 - \alpha_i(t-1))\hat{\mathbf{x}}^i_{t-1,B}$

4: **else**

5:     $\hat{\mathbf{x}}^{I_i}_{t,B} = \hat{\mathbf{x}}^i_{t-1,B}$

6: **end if**

7: Broadcast $\{t, i, d_{t,i}, \hat{\mathbf{x}}^{I_i}_{t,B}, \hat{\mathbf{x}}^i_{t-1,B}, \hat{w}_{ii}(t-1)\}$ and receive $\{t, j, d_{t,j}, \hat{\mathbf{x}}^{I_j}_{t,B}, \hat{\mathbf{x}}^j_{t-1,B}, \hat{w}_{jj}(t-1)\}$ from the communicating neighboring robots $j \in \Omega_{t,i}$

8: $w_{ij}(t-1) = \frac{\hat{w}_{jj}(t-1)}{\sum_{\forall j' \in \Lambda_{t,i}} \hat{w}_{j'j'}(t-1)}$

9: $\hat{\mathbf{x}}^i_{t,B} = \sum_{\forall j \in \Lambda_{t,i}} w_{ij}(t-1)\hat{\mathbf{x}}^{I_j}_{t,B}$, where $\Lambda_{t,i} = \Omega_{t,i} \cup \{i\}$

10: $\Delta\alpha_i(t-1) = \alpha_i(t-1) - \alpha_i(t-2)$; $\Delta d_i(t) = d_{t,i} - d_{t-1,i}$

11: *Periodic reset*: set the weights $\hat{\alpha}_i(t-1)$, $\hat{\alpha}'_i(t-1)$, and $\hat{w}_{ii}(t-1)$ to 1 after every $T_p$ discrete time steps

12: *Periodic perturbation*: after every $T_p$ discrete time steps, $e_p$ either takes the value $e_p = p_{mag}$ or $e_p = 0$ with equal probability, whereas $e'_p = p_{mag} - e_p$; otherwise, $e_p = e'_p = 0$ at all other times

13: $r^\alpha_{t,i} = e_{a1}\Delta\alpha_i(t-1)\Delta d_i(t) + e_p(1 - d_{t,i}) + e_{a2}\alpha_i(t-1)d_{t,i}$

14: $r^{\alpha'}_{t,i} = -e_{a1}\Delta\alpha_i(t-1)\Delta d_i(t) + e'_p(1 - d_{t,i}) + e_{a2}(1 - \alpha_i(t-1))d_{t,i}$

15: $\hat{\alpha}_i(t) = \hat{\alpha}_i(t-1)\exp\left(r^\alpha_{t,i}\right)$; $\hat{\alpha}'_i(t) = \hat{\alpha}'_i(t-1)\exp\left(r^{\alpha'}_{t,i}\right)$

16: $\hat{w}_{ii}(t) = \hat{w}_{ii}(t-1)\exp\left(-\eta_w(1 - d_{t,i})\right)$

---

---

**Algorithm 3** AOL-2P (for the $i^{th}$ robot, $i \in [N]$)

---

    **Choose:** $D_o > 0$, $\eta_\alpha > 0$, $\eta_w > 0$, $T_p \geq 1$ (integer-valued), $e_{a1} > 0$, $e_{a2} \geq 0$, $p_{mag} > 0$, $e_{w1} > 0$, $e_{w2} \geq 0$

    **Initialize:** $\hat{\alpha}_i(0) = \hat{\alpha}'_i(0) = \hat{\alpha}_i(-1) = \hat{\alpha}'_i(-1) = 1$, $w_{ii}(0) = w_{ii}(-1) = 1$, $\hat{\mathbf{x}}^i_{0,B} = \hat{\mathbf{x}}^{S_i}_{0,B}$

    **Input:** $t$, $d_{t,i}$, $\hat{\alpha}_i(t-1)$, $\hat{\alpha}'_i(t-1)$, $\hat{w}_{ii}(t-1)$, $\hat{\mathbf{x}}^{S_i}_{t,B}$, $\hat{\mathbf{x}}^i_{t-1,B}$, $d_{t-1,i}$, $\alpha_i(t-2)$, $w_{ii}(t-2)$

    **Output:** $\hat{\alpha}_i(t)$, $\hat{\alpha}'_i(t)$, $\hat{w}_{ii}(t)$, $\hat{\mathbf{x}}^i_{t,B}$

---

1: $\alpha_i(t-1) = \frac{\hat{\alpha}_i(t-1)}{\hat{\alpha}_i(t-1) + \hat{\alpha}'_i(t-1)}$

2: **if** $d_{t,i} > 0$ **then**

3:     $\hat{\mathbf{x}}^{I_i}_{t,B} = \alpha_i(t-1)\hat{\mathbf{x}}^{S_i}_{t,B} + (1 - \alpha_i(t-1))\hat{\mathbf{x}}^i_{t-1,B}$

4: **else**

5:     $\hat{\mathbf{x}}^{I_i}_{t,B} = \hat{\mathbf{x}}^i_{t-1,B}$

6: **end if**

7: Broadcast $\{t, i, d_{t,i}, \hat{\mathbf{x}}^{I_i}_{t,B}, \hat{\mathbf{x}}^i_{t-1,B}, \hat{w}_{ii}(t-1)\}$ and receive $\{t, j, d_{t,j}, \hat{\mathbf{x}}^{I_j}_{t,B}, \hat{\mathbf{x}}^j_{t-1,B}, \hat{w}_{jj}(t-1)\}$ from the communicating neighboring robots $j \in \Omega_{t,i}$

8: $w_{ij}(t-1) = \frac{\hat{w}_{jj}(t-1)}{\sum_{\forall j' \in \Lambda_{t,i}} \hat{w}_{j'j'}(t-1)}$

9: $\hat{\mathbf{x}}^i_{t,B} = \sum_{\forall j \in \Lambda_{t,i}} w_{ij}(t-1)\hat{\mathbf{x}}^{I_j}_{t,B}$, where $\Lambda_{t,i} = \Omega_{t,i} \cup \{i\}$

10: $\Delta\alpha_i(t-1) = \alpha_i(t-1) - \alpha_i(t-2)$

11: $\Delta w_{ii}(t-1) = w_{ii}(t-1) - w_{ii}(t-2)$; $\Delta d_i(t) = d_{t,i} - d_{t-1,i}$

12: *Periodic reset*: set the weights $\hat{\alpha}_i(t-1)$, $\hat{\alpha}'_i(t-1)$, and $\hat{w}_{ii}(t-1)$ to 1 after every $T_p$ discrete time steps

13: *Periodic perturbation*: after every $T_p$ discrete time steps, $e_p$ either takes the value $e_p = p_{mag}$ or $e_p = 0$ with equal probability, whereas $e'_p = p_{mag} - e_p$; otherwise, $e_p = e'_p = 0$ at all other times

14: $r^\alpha_{t,i} = e_{a1}\Delta\alpha_i(t-1)\Delta d_i(t) + e_p(1 - d_{t,i}) + e_{a2}\alpha_i(t-1)d_{t,i}$

15: $r^{\alpha'}_{t,i} = -e_{a1}\Delta\alpha_i(t-1)\Delta d_i(t) + e'_p(1 - d_{t,i}) + e_{a2}(1 - \alpha_i(t-1))d_{t,i}$

16: $\hat{\alpha}_i(t) = \hat{\alpha}_i(t-1)\exp(r^\alpha_{t,i})$; $\hat{\alpha}'_i(t) = \hat{\alpha}'_i(t-1)\exp(r^{\alpha'}_{t,i})$

17: *Periodic perturbation*: after every $T_p$ discrete time steps, $e^{w_i}_p = Unif.(0, p_{mag})$; otherwise, $e^{w_i}_p = 0$ at all other times; $Unif.(0, p_{mag})$ is a uniform random variable

18: $r^w_{t,i} = e_{w1}\Delta w_{ii}(t-1)\Delta d_i(t) + e^{w_i}_p(1 - d_{t,i}) + e_{w2}w_{ii}(t-1)d_{t,i}$

19: $\hat{w}_{ii}(t) = \hat{w}_{ii}(t-1)\exp(r^w_{t,i})$

---

