# OpenReview forum: "Reward-based Autonomous Online Learning Framework for Resilient Cooperative Target Monitoring using a Swarm of Robots"
_TMLR — Accepted by TMLR_

### Review · Reviewer_SbGK · 2024-04-15

**Summary Of Contributions:**

The paper considers a setting in which a swarm of robots has to follow and monitor a moving target. The robots have sensors that may experience failure (or occlusion), and they have local communication with limited bandwidth. The paper introduces a way for the robots to tune how they combine information received from their neighbors, by changing weights given to their neighbors according to some reward-based formula.

The proposed algorithm is analyzed theoretically, and numerical experiments show that it outperforms several baselines (albeit with high variance).

**Audience:**

Yes

**Broader Impact Concerns:**

The broader impact of this paper is not discussed in the paper. There does not seem to be a critical need for it, but for this particular paper, very applied and focusing on a sensitive topic (target tracking), I think that a broader impact statement would be welcome. How could this research be used for fugitive or political dissident tracking? Could the target be something else than a moving physical thing (for instance, would the equations still work to track some activity on social networks?)

**Claims And Evidence:**

Yes

**Requested Changes:**

Given the wide scope of the TMLR journal, I believe that clarity is of utmost importance, and that any work has to be precisely positioned with regards to related work, and the general field it considers. As such, my requested changes focus on clarity.

- Add some concise (e.g. a table) comparison of the proposed algorithms and some selected related work. This will also help the readers understand the novelty of the paper and the importance of the reward-based equations.
- One or all of the proposed methods should also have a concise pseudocode representation somewhere, for ease of reproducibility.
- Figure 3 is too big and has too much white space in it. It should be truncated, and the room feed by this operation used to show a nice system diagram of the proposed method, with a depiction of the target to track (what is known about it, what sensors produce that information, what the failure rate is), and some of the robots (with information about what sensors they have, what they observe about the target, how they communicate, what they communicate). This will be a repetition of the contents of the paper, but will greatly help give an overview of the proposed method. The cherry on the cake would be to highlight in that systems diagram the parts that are novel in this paper (for instance the learning equations), and have, in the caption, a one-line motivation of why that contribution really makes the whole system better.

**Strengths And Weaknesses:**

Strengths:

- The literature review seems complete and does a good job at introducing the field of cooperative target tracking
- The proposed algorithm is simple yet provides encouraging results in a challenging setting (with a high probability of link or sensor failure)

Weaknesses:

- The overal clarity of the paper is quite low, and it is difficult to quickly find what the contribution is, and what are the differences between AOL-C, AOL-1P and AOL-2P. It would help to have a table that compares a selection of related approaches with the 3 proposed algorithms (algorithm name in the rows, "features" in the columns such as "perturbation-greedy reward function").
- The paper seems on the edge of the scope of this journal, with its "learning" being the tuning of weights for an averaging operation, and the paper being very applied (it precisely focuses on one particular task for one particular set of robots, and even describes how the robots move, instead of focusing on the general problem of winding what neighbors to listen to to maximize some coverage or tracking objective).

---

> ### Author Response · Authors · 2024-04-29
> **Authors' response**
>
> The authors thank the reviewer for providing valuable insights to help improve the paper. As suggested by the reviewer, the authors will add a comparison table (comparing the AOL variants with related algorithms), add pseudocode for the three AOL variants, truncate Figure 3, and add a systems diagram summarizing the practical problem setting and how and where the AOL framework fits in the big picture.
>
> Further, the authors would like to address one of the weaknesses as pointed out by the reviewer regarding the generalizability of the AOL framework, as follows:
>
> In the paper, although the AOL framework has been demonstrated by implementing it in a practical setting involving cooperative target monitoring, the framework (and its theoretical analysis) by itself is quite general; the idea can be implemented in any such multi-agent problem setting or application where:
>
> 1) The agents have a shared environment, a common objective, and a common information space (estimation/prediction space).
> 2) An agent’s performance is governed by a reward-like signal.
> 3) How the agents fuse information from their neighbors (i.e., which neighbor to listen to) influences their performance or reward – accurate or quality information leads to better reward.
>
> The proposed AOL framework can be viewed as a reward-driven, online learning-based, adaptive information fusion framework that can be used by agents in a multi-agent problem setting to improve their reward/performance. Therefore, the AOL framework is useful in problems where the reward/performance of each agent is dependent on the quality/accuracy of the information that the agent has – the dependence can be direct (the fused information is directly responsible for reward generation) or indirect (the fused information is used to make a decision which is then responsible for reward generation).
>
> The problem of cooperative target monitoring considered in this paper involves an indirect dependence of the reward (target detection confidence) on the fused information (target position estimates). The reward-like signal, called target detection confidence or score, is produced by each robot’s exteroceptive sensor suite. The communicating robots utilize the AOL framework to determine what information should be given more weight in the weighted information fusion that improves their cumulative target detection score (cumulative reward). Therefore, the AOL framework (along with its theoretical analysis) works for any particular set of tasks, robots, or robot dynamics as long as the above-mentioned general conditions are satisfied. Examples include monitoring and controlling the spread of fire (target) in a forest, wildlife monitoring, search and rescue, etc.
>
> To clarify the generalizability of the AOL framework, the authors would like to add relevant points from the above discussion to section 3 of the paper, where the AOL framework has been introduced. Further, the authors would like to add an abstract diagram showing how the AOL framework can be applied in a general problem setting that follows the above discussion.
>
> Based on the reviewer’s advice, the authors have also prepared a broader impact statement relevant to the research in the paper.
>
> Broader impact statement:
> The reward-based autonomous online learning framework proposed in this paper called the AOL framework, is an adaptive information fusion framework that can be applied to various reward-based multi-agent applications, where the agents are required to be smart about which neighbors should be trusted when for a particular type of information, to increase their rewards as well as help others in the communication network.
> The problem considered in this paper, i.e., target search, tracking, and monitoring, can be used for many applications, like search and rescue, firefighting, disaster relief, convoy protection, wildlife monitoring, surveillance, etc. Moreover, the AOL framework can be useful in the search and detection of societal threats on the web by the use of multiple collaborative bots – the weights involved in the information fusion indicate which neighboring bot is detecting an increased threat level, thus helping in pinpointing the source of the activity in the overall network. Further, AOL can also be used in e-commerce applications, such as trading bots collaborating to increase profits, recommendation bots collaborating to improve customer reviews, etc.

---

### Review · Reviewer_MuFk · 2024-04-18

**Summary Of Contributions:**

The paper discusses a target monitoring task. Each robot has a narrow FOV, and a certain range at which it can communicate with nearby robots. These communications all have some probability $p$ of failing even if in range. Each robot follows 2 modes, object searching and object tracking.

Based on this setting, an online learning algorithm is proposed to optimize tracking. The reward used is the detection confidence
$d$, modeled after camera / LiDAR and increasing as distance approaches 0. All the approaches involve each robot maintaining a local estimate of the target's location, broadcasting that local estimate to nearby robots, and aggregating those estimates into a final estimate of position.

Much weight is placed upon the perturbation-greedy reward, which introduces some random noise into the reward estimate to encourage more exploration in target finding, scaled according to the models current confidence.

**Audience:**

Yes

**Broader Impact Concerns:**

The described task does feel very applicable to surveillance / military use cases. Although this is discussed in the introduction.

**Claims And Evidence:**

Yes

**Requested Changes:**

No changes

**Strengths And Weaknesses:**

Overall I think the approaches used to aggregate estimates are pretty reasonable. I appreciate that there is theoretical analysis of the convergence of the weights. I'm not sure how important the perturbations are to final performance - this sort of idea is pretty common in reinforcement learning style literature (see many works on optimizing under uncertainty). However the technicals of the paper seem good.

---

> ### Author Response · Authors · 2024-04-29
> **Authors' response**
>
> The authors thank the reviewer for highlighting the paper's contribution and its impact.
>
> Just like reinforcement learning uses exploration strategies to improve reward/performance, AOL-1P and AOL-2P use perturbations in their reward functions to explore their information fusion’s weights’ space for improving reward/performance.
>
> For the simulation setting used in the paper, for the number of robots equal to 5, 10, and 20, AOL-1P, the AOL variant that uses perturbations in its local learning layer, performs 3.62%, 9.41%, and 8.86% better in terms of cumulative detection score and 8.37%, 3.42%, and 4.5% better in terms of cumulative closeness score, respectively, than the variant AOL-C which does not use perturbations.

---

### Review · Reviewer_aS9w · 2024-10-21

**Summary Of Contributions:**

This paper considers the problem of monitoring a target with a swarm of robots. Two robots can communicate if they are close enough to each others, but the communication may fail with some probability. The problem is cooperative in nature. Three algorithms are proposed, based on Autonomous Online Learning (AOL). At a high level, the three algorithms have in common to involve a local estimation phase, a communication phase, a social estimation phase and then a learning phase. They differ in how the learning is implemented.  A convergence analysis as well as experimental results are provided, including a comparison with two existing baselines.

**Audience:**

Yes

**Broader Impact Concerns:**

A discussiong on the potential ethical implications (e.g., real-world tracking by robots) could be discussed.

**Claims And Evidence:**

Yes

**Requested Changes:**

1. Page 9, Section 4: Could you please clarify what you mean by "Without the loss of generality"? How would the assumptions and the results change with a periodic reset?

2. Page 13, Assumption 5: Below the assumption, you provide an example ("For instance ..."). But this example requires the i-th robot to be fixed over time. Does this mean in Theorem 3 that all the robots would need to be fixed over time? This seems overly restrictive.

**Strengths And Weaknesses:**

Strengths: (1) the paper addresses a problem which seems important for applications; (2) the paper contains a theoretical analysis

Weaknesses: (1) it is not clear to me how to check some of the assumptions in practical scenarios; (2) I am not sure how the proposed algorithm compares with state-of-the-art method for similar problems.

---

> ### Author Response · Authors · 2024-10-24
> **Authors' Response**
>
> The authors thank the reviewer for highlighting the paper’s contributions.
>
> The authors will first address the requested changes and then the weaknesses.
>
> The theoretical analysis of the three AOL variants without the periodic reset is carried out to gain insights into their convergence behavior just after a periodic reset is hit and just before the next periodic reset is about to hit. Note that each time instant when the periodic reset hits can be considered as $t = 0$ from an analysis perspective. Therefore, the analysis can be regarded as valid for each time interval between two successive periodic reset hits. Further, considering the limit $t \rightarrow \infty$  is a standard way of showing the exponential convergence behavior of a quantity or variable. In practice, given a periodic reset time $T_p$, a higher learning rate (set by learning rate parameters) would lead to quick convergence of the weights before the next periodic reset hits. If the learning rate is low, the weights may not converge to their respective convergence values by the time the periodic reset hits. However, their convergence behavior remains the same (exponential convergence), as shown in their theoretical analysis.
>
> The example discussed in remark 6 (under assumption 5) where the best (${q_{t,i}^*}^{th}$) robot is fixed over time is just one of the many possible scenarios that satisfy assumption 5. The example provided lies at one end of the spectrum ($β=1$) of the many possible scenarios encompassed by assumption 5. Thus, theorem 3 does not require the robots to be fixed over time; the example is just a special case of assumption 5. Another example satisfying theorem 3 can be the scenario where the best (${q_{t,i}^*}^{th}$) robot stays fixed for some finite time duration and may change intermittently over time.
>
> Note that the ${q_{t,i}^*}^{th}$ robot is defined (check page 12, section 4.3) as the robot for which a cumulative change in its social weight causes the most positively or least negatively aligned cumulative change in its detection confidence score compared to other robots in the neighborhood at time $t$. Further, note that the cumulative sum of the difference-based correction terms for the ${q_{t,i}^*}^{th}$ robot indicates (check page 9, perturbation-greedy reward function paragraph) whether an increase or decrease in its social weight increases or decreases the detection confidence score by how much.
>
> In essence, assumption 5 implies that this cumulative sum for the ${q_{t,i}^*}^{th}$ robot is more than that of other robots in its neighborhood by a finite term that grows linearly or sub-linearly in time $t$ but can be arbitrarily small in its growth rate ($0<β≪1$) and magnitude ($0<ϵ≪1$).  In practice, this would be satisfied for scenarios where the ${q_{t,i}^*}^{th}$ robot is fixed for some finite time duration and changes intermittently over time. In the simulation studies, such intermittent changes result from temporary/permanent sensor failures, intermittent communication link loss, and target visual loss.
>
> Assumptions 1 and 2 are required to show convergence of social weights involved in AOL-C and AOL-1P. An assumption similar to assumption 2 can be made to show the convergence of local weights involved in AOL-C following a similar analysis as shown for proving theorem 1. Assumption 3 is required to show convergence of local weights involved in AOL-1P and AOL-2P. Assumptions 4 and 5 are required to show convergence of social weights for AOL-2P.
>
> In practice, assumptions 1 and 4 imply that each robot’s neighborhood configuration (i.e., its neighboring robots) and performance configuration (i.e., the best-performing robot among itself and its neighbors) is fixed for some time duration. From an analysis perspective, considering the limit $t \rightarrow \infty$ allows us to analyze the exponential convergence characteristics of the weights. These assumptions are essential since learning cannot converge if the neighborhood and performance configurations keep on changing at every time step. Note that the analysis carried out without considering a periodic reset applies to the convergence behavior during the time between two successive periodic resets; therefore, the weights may converge to different values after every periodic reset depending upon the observed configuration until the next periodic reset hits.
>
> Assumptions 2, 3, and 5 support the theoretical analysis by giving a mathematical structure to the best-performing robot’s (or, in the case of assumption 3, the best-performing local estimate’s) performance compared to the other robots (or estimates), thereby making the analysis tractable for proving exponential convergence. In practice, these assumptions are satisfied if the neighborhood and performance configurations are fixed for some finite time duration and may change intermittently over time.

---

> ### Author Response · Authors · 2024-10-24
> **continuation...**
>
> The AOL framework is useful in problems where the reward/performance of each agent is dependent on the quality/accuracy of the information (estimates or predictions) that the agent has – the fused information can either be directly responsible for reward generation or be used to decide an action which in turn is responsible for reward generation. Similar problems where the AOL framework can be used are searching and detecting societal threats on the web by using multiple collaborative bots, trading bots collaborating to increase profits, recommendation bots collaborating to improve customer reviews, etc. For all such problems, the AOL algorithms are expected to perform better than other information fusion methods that utilize averaging consensus and Kalman filter-based techniques since the AOL framework involves an adaptive distributed multi-estimate fusion process driven/tuned online to improve performance based on a reward-like signal.
>
> The authors have also prepared a broader impact statement relevant to the research in the paper.
>
> Broader impact statement:
>
> The reward-based autonomous online learning framework proposed in this paper called the AOL framework, is an adaptive information fusion framework that can be applied to various reward-based multi-agent applications, where the agents are required to be smart about which neighbors should be trusted when for a particular type of information, to increase their rewards as well as help others in the communication network.
> The problem considered in this paper, i.e., target search, tracking, and monitoring, can be used for many applications, like search and rescue, firefighting, disaster relief, convoy protection, wildlife monitoring, surveillance, etc. Moreover, the AOL framework can be useful in the search and detection of societal threats on the web by the use of multiple collaborative bots – the weights involved in the information fusion indicate which neighboring bot is detecting an increased threat level, thus helping in pinpointing the source of the activity in the overall network. Further, AOL can also be used in e-commerce applications, such as trading bots collaborating to increase profits, recommendation bots collaborating to improve customer reviews, etc.

---

### Decision · Action_Editor_n2QF · 2024-11-27

**Recommendation:** Accept as is

**Comment:**

SbGK, after the rebuttal: "The authors increased the clarity of the paper and added a comparison against state-of-the-art baselines. The challenges tackled by the paper are now clear enough, and the contribution meaningful and generalizable."

SbGK, before the rebuttal: "The paper seems on the edge of the scope of this journal, with its "learning" being the tuning of weights for an averaging operation, and the paper being very applied (it precisely focuses on one particular task for one particular set of robots, and even describes how the robots move"

MuFk: "I appreciate that there is theoretical analysis of the convergence of the weights. I'm not sure how important the perturbations are to final performance"

**Audience:**

The paper is of interest to swarm robotics and swarm agents community, given the applied nature of it.

**Claims And Evidence:**

The paper addresses decentralized cooperative monitoring with a robot swarm under sensor failures. The authors propose three variants Autonomous Online Learning (AOL) with a reward and consensus-based weighted information fusion process.

The reviewers agree that the paper presents a reasonable approach albeit to a somewhat limited domain. The reviewers appreciate convergence  analysis and the addition of the baseline comparisons.

---

> ### Author Response · Authors · 2024-12-24
> **Authors' Comment**
>
> The authors thank the Action Editor for managing the review process and recognizing this work. The camera-ready version of the paper has been submitted, along with a link to the video presentation and the code.